# A consistent budgeting of terrestrial carbon fluxes

Lea Dorgeist [1], Clemens Schwingshackl [1] ✉, Selma Bultan [1] & Julia Pongratz[1,2]

Accurate estimates of $CO_2$ emissions from anthropogenic land-use change ($E_{LUC}$) and of the natural terrestrial $CO_2$ sink ($S_{LAND}$) are crucial to precisely know how much $CO_2$ can still be emitted to meet the goals of the Paris Agreement. In current carbon budgets, $E_{LUC}$ and $S_{LAND}$ stem from two model families that differ in how $CO_2$ fluxes are attributed to environmental and land-use changes, making their estimates conceptually inconsistent. Here we provide consistent estimates of $E_{LUC}$ and $S_{LAND}$ by integrating environmental effects on land carbon into a spatially explicit bookkeeping model. We find that state-of-the-art process-based models overestimate $S_{LAND}$ by 23% (min: 8%, max: 33%) in 2012–2021, as they include hypothetical sinks that in reality are lost through historical ecosystem degradation. Additionally, $E_{LUC}$ increases by 14% (8%, 23%) in 2012–2021 when considering environmental effects. Altogether, we find a weaker net land sink, which makes reaching carbon neutrality even more ambitious. These results highlight that a consistent estimation of terrestrial carbon fluxes is essential to assess the progress of net-zero emission commitments and the remaining carbon budget.

The anthropogenic usage of land and fossil fuels has massively altered the carbon balance of terrestrial ecosystems over the last decades, centuries, and even millennia[1,2]. An accurate knowledge of the terrestrial carbon budget is essential for estimating the fate of $CO_2$ emissions and, thus, for understanding past and projecting future climate change. The terrestrial carbon budget (Table 1) is composed of $CO_2$ fluxes due to anthropogenic land-use changes (e.g., deforestation, afforestation) and due to environmental changes on land (effects of rising $CO_2$ levels, climate change, and nitrogen deposition). The Global Carbon Project's annual Global Carbon Budget (GCB[1]) estimates that land use, land-use change and forestry (LULUCF) has been a net source of $CO_2$ throughout the industrial era and contributed 12% of total $CO_2$ emissions in 2013–2022. Environmental changes, in contrast, cause a net $CO_2$ uptake by terrestrial ecosystems in most years and offset 31% of total anthropogenic $CO_2$ emissions averaged over 2013–2022[1]. The GCB carbon flux estimates are of paramount relevance since important global reports on climate change and climate mitigation efforts, such as the assessment reports by the Intergovernmental Panel on Climate Change (IPCC)[3,4],

the United Nations Environment Programme (UNEP) gap report[5], and a report on indicators of global climate change[6], rely on them.

Currently, the GCB uses two different types of models to provide historical estimates of anthropogenic $CO_2$ emissions from LULUCF ($E_{LUC}$) and of the natural land sink ($S_{LAND}$)[1,5]. $E_{LUC}$ is estimated by semi-empirical, observation-driven bookkeeping models. Bookkeeping models calculate $E_{LUC}$ by combining the area affected by LULUCF with carbon densities of vegetation and soil and empirical growth and decay curves of carbon stored in vegetation, soil, and harvested wood products (see Methods). Key features of bookkeeping models are that they enable the separation of direct anthropogenic fluxes from natural fluxes on land and their traceability of $E_{LUC}$ to specific LULUCF events[7–10]. $S_{LAND}$ estimates stem from simulations of Dynamic Global Vegetation Models (DGVMs) that are conducted within the TRENDY model intercomparison project[11]. DGVMs are process-based carbon cycle models that simulate plant and soil processes in response to external environmental drivers, such as rising $CO_2$ levels and meteorological and climate variability[12,13].

[1]Department of Geography, Ludwig-Maximilians-Universität, München, Germany. [2]Max Planck Institute for Meteorology, Hamburg, Germany.
✉ e-mail: c.schwingshackl@lmu.de

**Table 1 | Terms of the terrestrial carbon budget**

| | |
|---|---|
| Terrestrial carbon budget | Entirety and balance of all carbon fluxes between land and atmosphere |
| LULUCF | Land use, land-use change and forestry |
| $E_{LUC}$ | Anthropogenic $CO_2$ emissions due to LULUCF. $E_{LUC,trans}$ includes transient environmental effects on $E_{LUC}$, $E_{LUC,pd}$ includes environmental effects on $E_{LUC}$ based on present-day environmental conditions, and $E_{LUC,pi}$ excludes all environmental effects. |
| Environmental contribution to $E_{LUC}$ ($\delta L$) | Effect of transient environmental changes on $E_{LUC}$ calculated as the difference between a simulation accounting for transient environmental changes ($E_{LUC,trans}$) and a simulation excluding environmental changes ($E_{LUC,pi}$). |
| $S_{LAND}$ | Natural land sink, which comprises the carbon fluxes due to environmental changes, such as rising $CO_2$ levels, climate change, and nitrogen deposition (but excludes $E_{LUC}$ and $\delta L$). $S_{LAND,pi}$ is calculated under pre-industrial land cover from 1700, and $S_{LAND,trans}$ is calculated on actual, transiently changing land cover. |
| Net land flux | Net carbon flux between land and atmosphere. It is the sum of $E_{LUC,trans}$ and $S_{LAND,trans}$. |
| Replaced sinks and sources (RSS) | RSS are the hypothetical carbon sinks in ecosystems lost due to ecosystem degradation. RSS also comprises gained sinks in case of ecosystem restoration. RSS is calculated as the difference between $S_{LAND,trans}$ and $S_{LAND,pi}$. |
| Loss of additional sink capacity (LASC) | LASC is the sum of RSS and $\delta L$. It comprises the effect of environmental changes on $S_{LAND}$ and $E_{LUC}$ on actual, transiently changing land cover (due to LULUCF) compared to a state with pre-industrial land cover from 1700 (note that some publications use the LASC term to refer only to the RSS term, e.g. ref. 9 and ref. 14). |

The sum of $E_{LUC}$ from bookkeeping models and $S_{LAND}$ from DGVMs does not reliably represent the terrestrial carbon budget due to fundamental differences in how these two model families account for environmental and land-use changes. Bookkeeping models typically use time-invariant carbon densities from inventories or models. By assuming a steady environmental state, they neglect environmental changes preceding or succeeding a LULUCF event (e.g., denser growing forests in response to a rising atmospheric $CO_2$ concentration, which emit more when cleared for agricultural land[14]). Consequently, bookkeeping models estimate larger (smaller) $E_{LUC}$ values for all years preceding (succeeding) the date of origin of the carbon densities. In contrast, DGVMs consider transient environmental effects (effects changing over time) for estimating $S_{LAND}$. Historically, environmental effects, such as rising $CO_2$ levels, have been mainly beneficial for plant growth, in particular for forests with their long-lived woody biomass. As a consequence, $S_{LAND}$ has been a carbon sink globally[1]. However, by design of the simulation setup, DGVMs estimate $S_{LAND}$ under pre-industrial land cover, thus including effects of environmental changes in forest areas that, in reality, have since been lost due to LULUCF. The hypothetical carbon sinks in these lost ecosystems are also known as replaced sinks and sources[15] (RSS). The RSS term is of substantial size with 31 GtC of hypothetical sinks cumulatively from 1850–2018 (as estimated by ref. 9 with a bookkeeping method). Due to limitations in the simulation setup, RSS cannot be isolated directly with DGVMs (see Methods). In analyses based on DGVMs, RSS are always lumped together with the effect of environmental changes on $E_{LUC}$ in a term summarized as loss of additional sink capacity[16] (LASC). The LASC term combines carbon fluxes from environmental changes on land that have been altered due to LULUCF and from changes in $E_{LUC}$ due to environmental effects (see Table 1). The missing environmental effects in the bookkeeping estimates of $E_{LUC}$ and the assumption of a constant, pre-industrial land cover in the DGVM estimates of $S_{LAND}$ currently prohibit closing the terrestrial carbon budget.

In this study, we present an approach to overcome the current inconsistencies within the terrestrial carbon budget. We derive transient carbon densities for vegetation and soil from DGVMs (responding to changes in environmental conditions) and implement them into the Bookkeeping of Land Use Emissions model[7] (BLUE), one of three bookkeeping models routinely used in the GCB to deliver $E_{LUC}$ estimates[1]. This method enables BLUE to capture transient environmental conditions while keeping the traceability and flexibility of the bookkeeping approach (Supplementary Fig. 1). The capacity of BLUE to simulate spatially explicit carbon fluxes at 0.25° resolution allows us to identify the hotspots of human interference with natural ecosystems. We advance current $E_{LUC}$ estimates that neglect environmental effects on carbon stocks by providing an $E_{LUC}$ estimate for the actual

emissions released into the atmosphere upon a LULUCF event, making our estimates politically more relevant. Additionally, we provide an estimate of $S_{LAND}$ under transient land cover instead of pre-industrial land cover. Our bookkeeping estimate of $S_{LAND}$ thus excludes the lost sinks due to historical ecosystem degradation and is therefore conceptually much closer to reality than the current GCB estimate. Lastly, we quantify RSS and LASC to unveil the biases introduced by considering a pre-industrial instead of the transient land cover for $S_{LAND}$ and by neglecting the effects of environmental changes on $E_{LUC}$. Our study therefore offers three fundamental advances: (i) We propose an approach for a consistent terrestrial carbon budget, without omission or double counting of fluxes, which attributes fluxes intuitively to natural or anthropogenic drivers. (ii) Our developments enable the spatially explicit bookkeeping model BLUE to simulate the complete (natural and anthropogenic) terrestrial carbon balance. (iii) We provide state-of-the-art estimates of all terrestrial budget fluxes, including previously often ignored fluxes like lost sinks from ecosystem degradation.

## Results
### Effect of environmental changes on emissions from land-use change
Including the effect of transient environmental changes on $E_{LUC}$ ($E_{LUC,trans}$) increases annual emissions compared to $E_{LUC}$ calculated under pre-industrial environmental conditions ($E_{LUC,pi}$) by 28% (21%, 38%; range across five estimates, see Methods) averaged over 2012–2021. This corresponds to 0.34 (0.18, 0.56) GtC yr$^{-1}$ higher emissions compared to $E_{LUC,pi}$ (Fig. 1a, Table 2). Cumulative emissions in 1850–2021 increase by 11% (10%, 12%), corresponding to 25 (14, 42) GtC. The higher values of $E_{LUC,trans}$ are related to enhanced carbon uptake in vegetation and soil in response to a rising atmospheric $CO_2$ concentration and other potentially favorable environmental effects (e.g., nitrogen deposition; see Supplementary Fig. 2 for a map of $E_{LUC,trans}$). This is reflected in the increasing carbon densities within vegetation and soil for most biomes[17–19] (Supplementary Figs. 3 and 4). Upon deforestation and wood harvest, this additional carbon is released and causes larger values of $E_{LUC,trans}$ compared to $E_{LUC,pi}$. Assuming present-day environmental conditions for the whole simulation period ($E_{LUC,pd}$), which is the current GCB approach, overestimates (underestimates) $E_{LUC}$ before (after) the date of the inventory-based carbon densities (around 1980; see Methods). In 2012–2021, $E_{LUC,trans}$ therefore exceeds $E_{LUC,pd}$ by 14% (8%, 23%; Table 2).

Upon deforestation and wood harvest, the higher carbon stocks of vegetation and soil increase $CO_2$ emissions in $E_{LUC,trans}$ compared to $E_{LUC,pi}$ by 24% (0.4 GtC yr$^{-1}$) and 22% (0.3 GtC yr$^{-1}$, Fig. 1b,

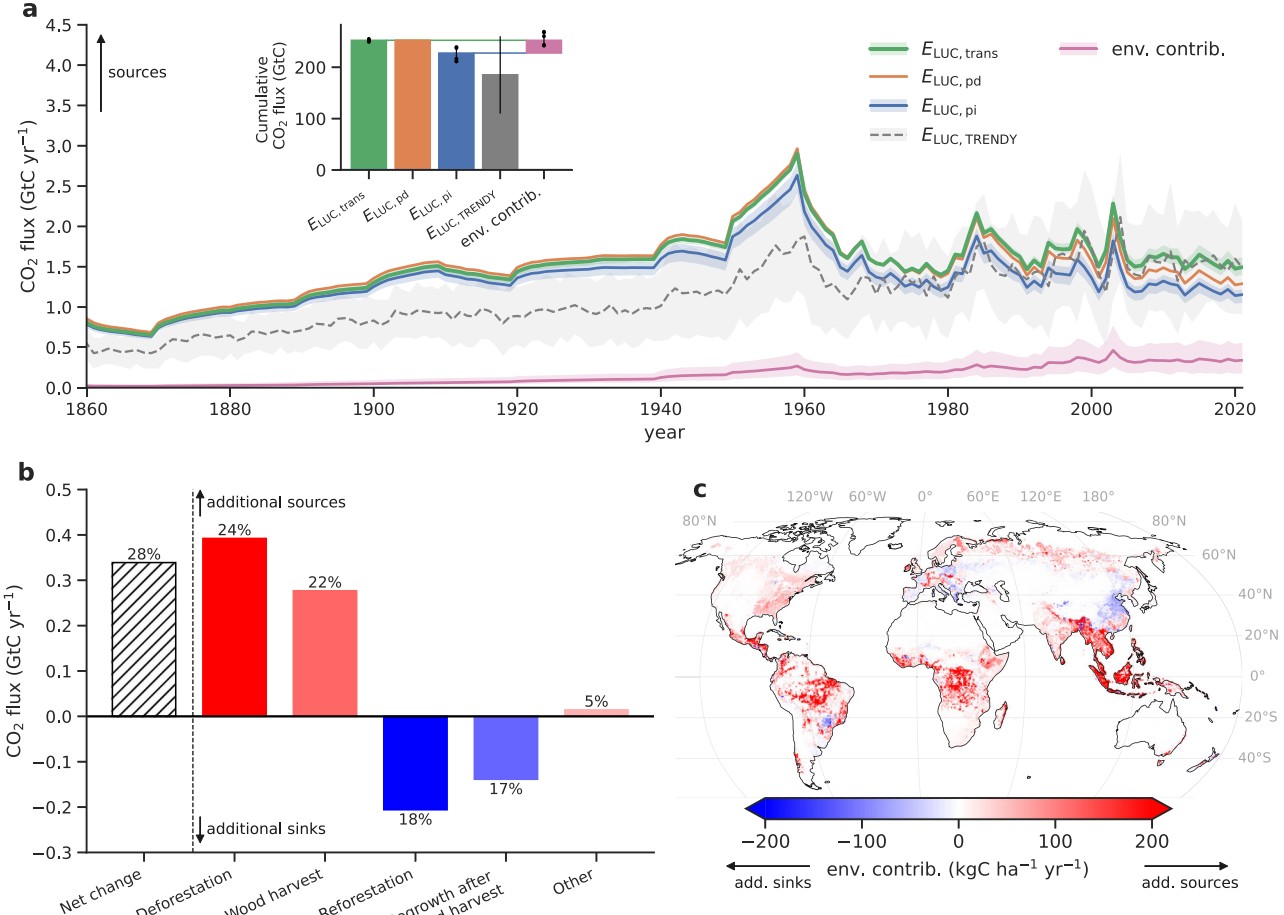

**Fig. 1 | Spatiotemporal effects of different environmental conditions on land-use change emissions ($E_{LUC}$). a** Comparison of global annual $E_{LUC}$ estimates from three simulations with the bookkeeping model BLUE applying transient ($E_{LUC,trans}$), present-day ($E_{LUC,pd}$), and pre-industrial ($E_{LUC,pi}$) carbon densities and from Dynamic Global Vegetation Models (DGVMs) from the TRENDY project ($E_{LUC,TRENDY}$). $E_{LUC,trans} - E_{LUC,pi}$ yields the environmental contribution (env. contrib.) to $E_{LUC}$, i.e., the additional sinks and sources due to environmental effects. The inset in **a** shows the corresponding global cumulative values (1850–2021). Uncertainties (shaded areas in time series and whiskers in inset) indicate the minimum-to-maximum range across BLUE estimates obtained by scaling BLUE carbon densities with individual DGVMs (dots indicate individual estimates; see Methods) and one standard deviation for TRENDY estimates. Our BLUE estimates are based on simulations using averaged DGVM carbon densities (see Methods); thus, they do not correspond to the mean of the BLUE estimates based on individual DGVMs. **b** Environmental contribution to $E_{LUC}$ of major land-use transitions and land-management types averaged over 2012–2021 as absolute values and as percentage change of each component relative to pre-industrial conditions. **c** Spatial distribution of the environmental contribution to $E_{LUC}$ averaged over 2012–2021. The map of $E_{LUC,trans}$ as our suggested most comprehensive estimate of $E_{LUC}$ can be found in Supplementary Fig. 2. Source data are provided as a Source Data file.

Supplementary Figs. 5 and 6). These larger emissions are only partly compensated by increased sinks through re/afforestation (increase by 18%, 0.2 GtC yr$^{-1}$) and regrowth after wood harvest (increase by 17%, 0.1 GtC yr$^{-1}$). Environmental impacts on $E_{LUC}$ are largest in tropical regions (Fig. 1c, Supplementary Fig. 7), where relatively recent and substantial forest clearings occurred under strongly increased carbon stocks. Southeast Asia, Equatorial Africa, and Brazil mainly contribute to the global increase in emissions with 27%, 19%, and 18%, respectively. By contrast, the (smaller) areas of re/afforestation, as in Europe or China, provide a larger sink than would be estimated without considering environmental changes (Fig. 1c).

Our results are in line with findings from ref. 9, who estimated a 19% increase in $E_{LUC}$ in 2009–2018 (Table 2, Supplementary Table 1) when applying transient instead of pre-industrial environmental conditions by emulating transient DGVM carbon densities in the Earth System Simulator OSCAR. The OSCAR estimate might be more conservative than ours, as the DGVM carbon densities used by OSCAR are lower than our BLUE carbon densities (Supplementary Fig. 8). Additionally, they are aggregated into biomes without distinguishing between primary and secondary land[9], a distinction which is considered in BLUE (see Methods). TRENDY models also deliver a transient estimate of $E_{LUC}$ (shown in Fig. 1a), which is however conceptually not comparable to $E_{LUC,trans}$, as the transient $E_{LUC}$ estimate from TRENDY (which is also reported by ref. 16) includes the RSS term (see Methods).

### Effect of transient land cover on the natural land sink

Our approach allows us to quantify $S_{LAND}$ under transient land cover ($S_{LAND,trans}$), which globally averages at −3.0 (−3.9, −2.2) GtC yr$^{-1}$ in 2012–2021 and amounts to a cumulative sink of −225 (−296, −160) GtC in 1850–2021 (Fig. 2, Table 2). These values are in good agreement with estimates from ref. 9. $S_{LAND,trans}$ is substantially lower (i.e., a weaker sink) than $S_{LAND}$ estimated with the conventional approach under pre-industrial land cover $S_{LAND,pi}$, which yields a sink of −3.7 (−2.6, −5.2) GtC yr$^{-1}$ in 2012–2021. $S_{LAND,trans}$ excludes the purely hypothetical $CO_2$ fluxes that would exist in ecosystems that in reality were lost due to LULUCF. The $S_{LAND}$ estimate of the GCB (−3.1 ± 0.9 GtC yr$^{-1}$ (mean and 1 SD) in 2021–2021; based on TRENDY models), which is calculated under pre-industrial land cover, erroneously includes these lost sinks (i.e., the replaced sinks and sources, RSS). Globally, we find a reduction

**Table 2 | Environmental and land-use components of the terrestrial carbon budget**

| | | $E_{LUC}$ | | | $S_{LAND}$ | | Net land flux (including peat) | LASC | RSS | Env. contrib. (δL) |
|---|---|---|---|---|---|---|---|---|---|---|
| | | Transient | Pre-industrial | Present-day | Transient | Pre-industrial | | | | |
| Mean (GtC yr⁻¹) | This study (2012–2021) | 1.5 [1.5, 1.7] | 1.2 [1.1, 1.3] | 1.4 | -3.0 [-3.9, -2.2] | -3.7 [-5.2, -2.6] | -1.2 [-2.1, -0.5] | 1.0 [0.5, 1.7] | 0.7 [0.3, 1.3] | 0.3 [0.2, 0.6] |
| | TRENDY (2012–2021) | 1.5±0.6 | - | - | - | -3.1±0.9 | -1.4±0.7 | - | - | - |
| | GCB2022[5] (2012–2021) | - | - | 1.0±0.4 | - | -3.1±0.9 | -1.9±1.0 | - | - | - |
| | Obermeier et al.[16] (2009–2018) | 2.0±0.6 | 1.2±0.5 | 1.6±0.8 | - | - | - | 0.8±0.3 | - | - |
| | Gasser et al.[9] (2009–2018) | 1.4±0.4 | - | - | -3.0±1.0 | - | -1.6±0.8 | - | 0.7±0.6 | - |
| | O₂ estimates[ ] (2013–2022) | - | - | - | - | - | -1.2±0.8 | - | - | - |
| | Atmospheric inversions[ ] (2012–2021) | - | - | - | - | - | -1.4 [-2.0, -0.3] | - | - | - |
| Cumulative (GtC) | This study (1850–2021) | 253 [249, 255] | 228 [212, 239] | 253 | -225 [-296, -160] | -258 [-358, -182] | 53 [-21, 117] | 58 [24, 95] | 33 [8, 62] | 25 [14, 42] |
| | TRENDY (1850–2021) | 185±75 | - | - | - | -212±113 | -2±112 | - | - | - |
| | GCB2022[5] (1850–2021) | - | - | 178±73 | - | -212±113 | -9±137 | - | - | - |
| | Obermeier et al.[16] (1850–2018) | 189±56 | 149±47 | 192±64 | - | - | - | 40±15 | - | - |
| | Gasser et al.[9] (1850–2018) | 178±50 | - | - | -205±53 | - | -27±26 | - | 31±22 | - |

Global decadal means in GtC yr⁻¹ and cumulative sums in GtC of land-use change emissions $E_{LUC}$ (calculated under transient, pre-industrial, and present-day environmental conditions), the natural land sink $S_{LAND}$ (calculated under the actual, transient and pre-industrial land cover), net land flux, the loss of additional sink capacity (LASC), the replaced sinks and sources (RSS), and the environmental contribution to $E_{LUC}$ (δL). Definitions of the individual terms are given in Table 1. Estimates stem from the bookkeeping model BLUE (this study). Dynamic Global Vegetation Models (DGVMs) from the TRENDY project, the Global Carbon Budget 2022 (GCB2022[5]), ref. 16, ref. 9, and ref. 1. GCB2022 calculates $E_{LUC}$ as the mean of three bookkeeping models (OSCAR, H&C23, and the $E_{LUC,PD}$ estimate of BLUE.) and $S_{LAND}$ from TRENDY simulations under pre-industrial land cover (see Methods). Since TRENDY is used in the GCB2022, $S_{LAND}$ estimates are identical in the TRENDY approach. The net land flux estimates of BLUE, TRENDY, and GCB2022 include emissions from peat fires and peat drainage from external datasets (see Fig. 4 and Methods; note that the net land flux estimate of ref. 9 does not include peat emissions). Peat emissions are 0.23 GtC yr⁻¹ averaged over 2012–2021 and cumulate to 25 GtC in 1850–2021. Uncertainties for BLUE are reported as minimum-to-maximum range across estimates obtained by scaling BLUE carbon densities with individual DGVMs, as minimum-to-maximum range across eight inversions for the atmospheric inversions estimate, and as one standard deviation for all other estimates. BLUE estimates (this study) for additional time periods can be found in Supplementary Table 1.

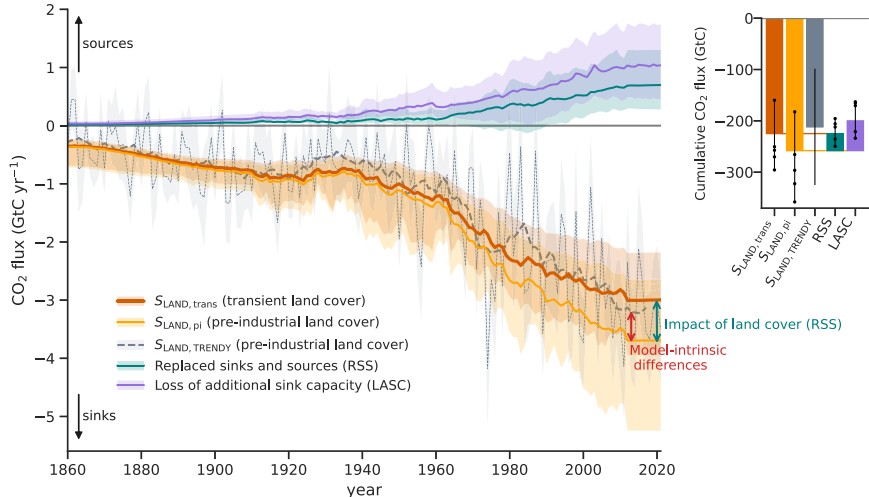

**Fig. 2 | The natural land sink ($S_{LAND}$) under different land cover assumptions and the resulting lost sinks.** Global averages of $S_{LAND}$ from the bookkeeping model BLUE under actual, transient land cover ($S_{LAND,trans}$) and under pre-industrial land cover ($S_{LAND,pi}$), and $S_{LAND}$ estimated with Dynamic Global Vegetation Models (DGVMs) from the TRENDY project ($S_{LAND,TRENDY}$; thin line indicates annual data and thick line indicates 10-year moving averages). The difference between $S_{LAND,trans}$ and $S_{LAND,pi}$ are the replaced sinks and sources (RSS), i.e., the lost (or gained) sinks due to degradation (or restoration) of ecosystems by land-use changes. The loss of additional sink capacity (LASC) is the sum of RSS and the environmental contribution to $E_{LUC}$ (see Fig. 1 and Table 1). $S_{LAND,pi}$ and $S_{LAND,TRENDY}$ are both based on pre-industrial land cover and, thus, their difference is due to model-intrinsic differences between BLUE and the DGVMs from the TRENDY project. The inset on the right shows the corresponding cumulative sums from 1850–2021. Uncertainties (shaded areas in time series and whiskers in inset) indicate the minimum-to-maximum range across BLUE estimates of $S_{LAND,trans}$, $S_{LAND,pi}$, RSS, and LASC obtained by scaling BLUE carbon densities with individual DGVMs (dots indicate individual estimates; see Methods) and one standard deviation for TRENDY estimates. Our BLUE estimates are based on simulations using averaged DGVM carbon densities (see Methods); thus, they do not correspond to the mean of the BLUE estimates based on individual DGVMs. Source data are provided as a Source Data file.

in $S_{LAND}$ due to LULUCF by 0.7 (0.3, 1.3) GtC yr$^{-1}$ in 2012–2021 and by 33 (8, 62) GtC cumulatively in 1850–2021 (RSS term in Fig. 2, Table 2). This corresponds to a 23% (8%, 33%) overestimation of the sink if assuming pre-industrial land cover in 2012–2021. LASC (RSS plus the environmental contribution to $E_{LUC}$; see Table 1) amounts to 1.0 (0.5, 1.7) GtC yr$^{-1}$ averaged over 2009–2018 (Supplementary Table 1) and thus accords with findings from ref. 16 (LASC of 0.8 ± 0.3 GtC yr$^{-1}$ in 2009–2018).

Regionally, RSS is largest in regions with a long history of ecosystem degradation, where favorable environmental effects could not accumulate over time (Fig. 3). Particularly, forest clearings over longer periods, such as in the eastern U.S. and eastern Europe, have increased RSS. Additionally, regions characterized by strong land-use disturbances in the last decades, such as many tropical forest areas in Latin America, Africa, and Southeast Asia, have started to contribute substantially to reducing sinks. We identify Brazil and southeast Asia (both mainly belonging to the tropical evergreen forest biome) as the main drivers of lost sinks globally (both having an RSS of 0.1 GtC yr$^{-1}$, respectively, Fig. 3c, Supplementary Fig. 9). This is mainly explained by high $E_{LUC}$ (Supplementary Fig. 7) and high observed carbon stock losses of biomass over history in both regions[20,21]. Moreover, we find the highest RSS in South Asia (38% lower sinks in $S_{LAND,trans}$ compared to $S_{LAND,pi}$, 2012–2021), southeast Asia (28%), Central America (27%), and China (25%, 2012–2021). In contrast, a gain in sinks is found for regions where reforestation occurred over the last two centuries, such as Central and Western Europe.

### Achieving a consistent estimate of the terrestrial carbon budget and its environmental and land-use components

Our consistent BLUE estimate of the net land flux as the sum of $E_{LUC,trans}$ and $S_{LAND,trans}$ suggests that land is a net sink of $CO_2$ amounting to −1.2 (−2.1, −0.5) GtC yr$^{-1}$ in 2012–2021 (Table 2, Fig. 4). This is a slightly weaker net sink than estimated by TRENDY (−1.4 ± 0.7 GtC yr$^{-1}$ in 2012–2021) and atmospheric inversion systems (−1.4 (−2.0, −0.3) GtC yr$^{-1}$ in 2012–2021), but similar to the net land flux estimate based on atmospheric $O_2$ observations (−1.2 ± 0.8 GtC yr$^{-1}$ in 2013–2022). The overall good agreement of these net land flux estimates reflects that they are conceptually comparable as they are all based on transient land cover and transient environmental conditions (see Methods). In contrast, the GCB estimate of the net land flux conceptually differs as it is the sum of the bookkeeping $E_{LUC}$ (resembling $E_{LUC,pd}$) and the TRENDY $S_{LAND}$ (resembling $S_{LAND,pi}$). Consequently, the GCB estimate yields a larger net sink of −1.9 ± 1.0 GtC yr$^{-1}$ in 2012–2021 than all the other approaches. This highlights that accounting for transient land cover and transient environmental conditions is crucial to accurately estimate $S_{LAND}$, $E_{LUC}$, and the net land flux and to reconcile existing approaches.

The cumulative net land flux over 1850–2021 from BLUE indicates that land has been a net source of $CO_2$ of 53 (−21, 117) GtC, while TRENDY estimates a small net $CO_2$ sink of −2 ± 112 GtC as does the GCB with −9 ± 137 GtC (Fig. 4, Table 2). The good agreement of the cumulative GCB estimate with TRENDY is likely due to compensating biases, with GCB having larger net emissions than TRENDY before 1970 and larger net sinks afterwards (Fig. 4). Another estimate of the net land flux based on atmospheric $CO_2$ and $\delta^{13}C$ records yields 31 (−26, 88) GtC cumulatively in 1850–1995[22]. For the same period, BLUE yields a cumulative net land flux of 78 (24, 123) GtC (Supplementary Table 1). As all these estimates bear large uncertainties, it remains inconclusive whether land has been a cumulative sink or source of $CO_2$ since 1850.

Comparisons of our estimates with further observation-based estimates of components of the terrestrial carbon budget are not easily possible. Observation-based datasets, including Earth observations and upscaled data from FLUXNET towers (FLUXCOM initiative[23]), cannot directly separate $S_{LAND}$ and $E_{LUC}$ without further knowledge and assumptions on the underlying drivers (i.e., anthropogenic or natural). Modeling approaches, like bookkeeping models or DGVMs, are thus essential to separate $S_{LAND}$ and $E_{LUC}$. Although Earth observations can provide estimates of the net land flux, they do not include all compartments of the terrestrial carbon cycle, e.g., studies focusing on

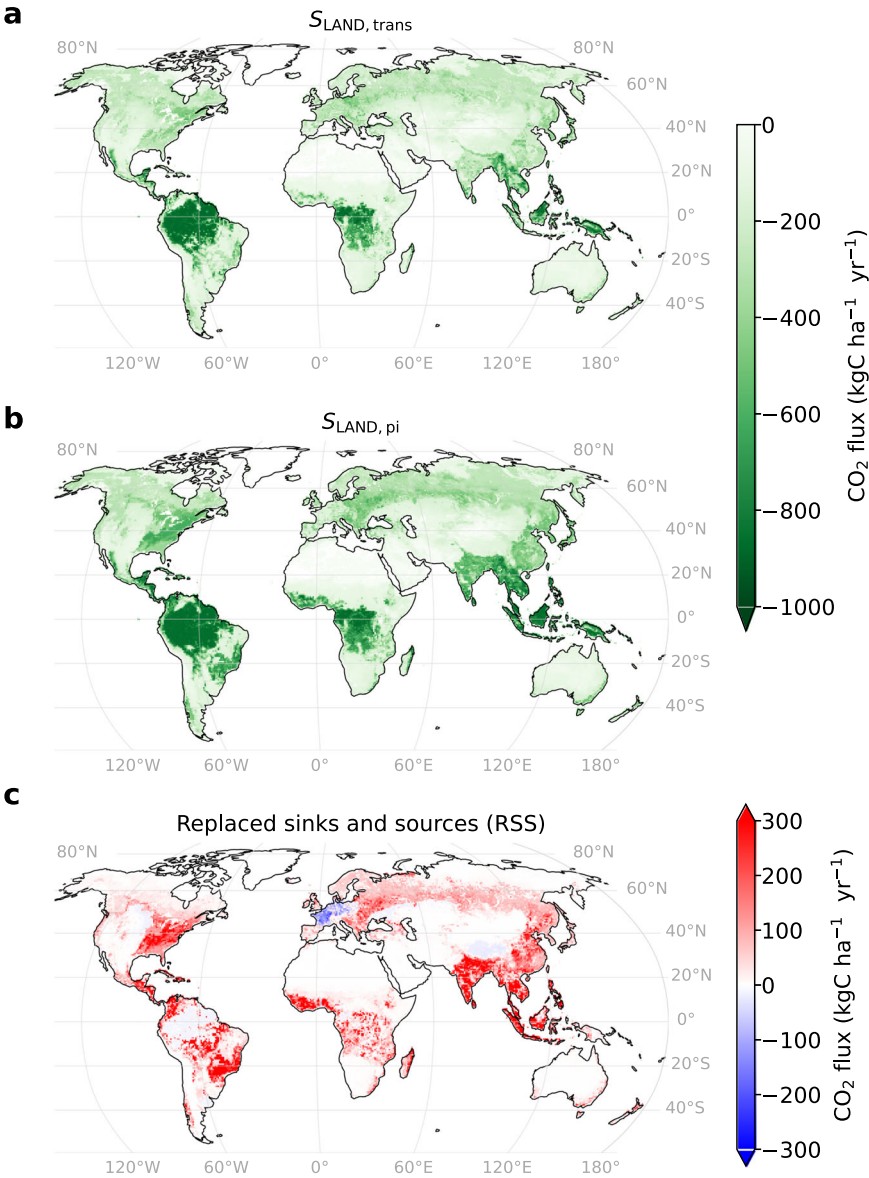

**Fig. 3 | Effects of different land-cover assumptions on the natural land sink.** Spatial distribution of the natural land sink ($S_{LAND}$) under (**a**) actual, transient land cover ($S_{LAND,trans}$) and (**b**) pre-industrial land cover ($S_{LAND,pi}$), averaged over 2012–2021. **c** Spatial distribution of replaced sinks and sources (RSS = $S_{LAND,trans} - S_{LAND,pi}$), i.e., the lost (or gained) sinks due to historical degradation (or restoration) of ecosystems. Source data are provided as a Source Data file.

forests[24] miss carbon fluxes from non-forested regions and studies focusing on live biomass[25] miss carbon fluxes from soils.

The net land flux from TRENDY consistently closes the carbon budget as does our BLUE estimate (see Methods). However, compared to our BLUE approach, some DGVMs lack important land management processes or represent them in a simplistic way[1,26–29]. Moreover, the split of the net land flux from TRENDY into $E_{LUC}$ ($E_{LUC,TRENDY}$) and $S_{LAND}$ ($S_{LAND,TRENDY}$) is confounded by LASC. With the current setup for DGVM simulations it is not possible to split LASC into RSS and the environmental contribution to $E_{LUC}$ (see Methods), and DGVMs can thus not deliver all terms necessary for a holistic and consistent terrestrial carbon budget. The higher net sink in the GCB results both from a larger $S_{LAND}$ (by assuming a hypothetical pre-industrial land cover and thereby including RSS) and a lower $E_{LUC}$ estimate (as the three GCB bookkeeping estimates mostly use time-invariant carbon densities based on inventories and models[7–9]).

Resolving the inconsistencies within the terrestrial carbon budget using BLUE has important implications for the budget imbalance, i.e.,

the mismatch between the sum of the estimated fossil and LULUCF $CO_2$ emissions and the sum of land, ocean, and atmosphere sinks. The budget imbalance shifts from −0.4 GtC yr⁻¹ to +0.3 GtC yr⁻¹ (2012–2021, Fig. 5), in line with our findings that propose an overestimation of $S_{LAND}$ and an underestimation of $E_{LUC}$ in the GCB. We note that our improvements to the terrestrial carbon budget do not bring the budget imbalance to zero - this cannot be expected due to many (potentially compensating) errors that accumulate in the imbalance term as a consequence of uncertainties in each of the five budget terms. Our results instead suggest a positive budget imbalance, which means that the estimated carbon sources are larger than the estimated carbon sinks. This imbalance could be explained by biases in other terms of the carbon budget. Indeed, a recent discussion suggests, among other potential causes for the imbalance, that estimates of the ocean sink in the GCB would be larger if they were based on the observation-based estimates of fugacity of $CO_2$ instead of being based on global ocean biogeochemistry models, as is the current GCB approach[1]. A larger ocean sink is also supported by atmospheric inversion estimates[1] and

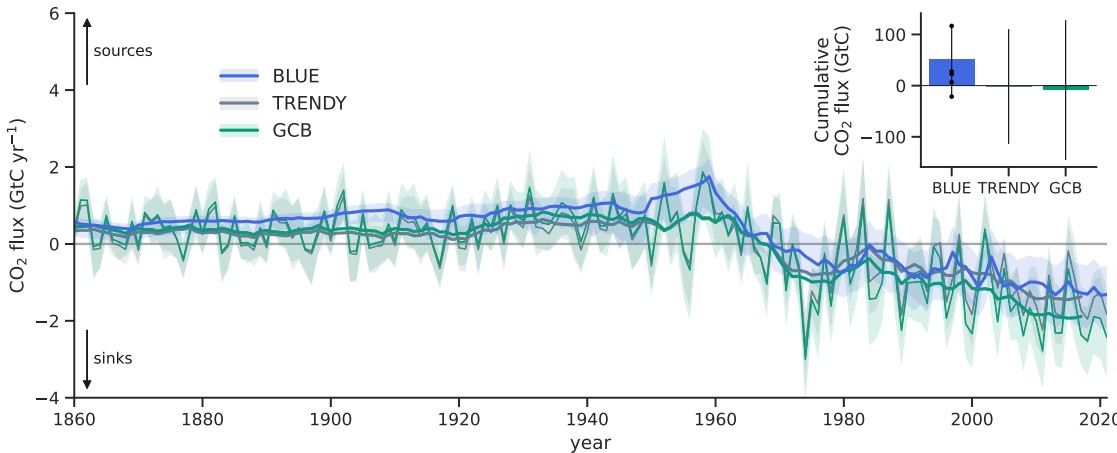

**Fig. 4 | Net land flux estimates from BLUE, TRENDY, and the GCB.** Time series of net land flux estimated with the bookkeeping model BLUE (considering transient environmental conditions and transient land-use change), Dynamic Global Vegetation Models (DGVMs) from the TRENDY project (based on the TRENDY S3 simulation, which uses transient environmental conditions and transient land-use change and is thus conceptually consistent with our BLUE estimate), and the Global Carbon Budget 2022 (GCB2022; sum of the natural land sink $S_{LAND}$ from TRENDY under pre-industrial land cover, resembling $S_{LAND,pi}$, and land-use change emissions $E_{LUC}$ from three bookkeeping models, resembling $E_{LUC,pd}$). See Methods for further details about the single estimation methods. Thin lines show annual values, thick lines show 10-year moving averages. The net land flux estimates include emissions from peat fires and peat drainage from external datasets (see Methods). The inset shows the corresponding cumulative values over 1850–2021. Uncertainties (shaded areas in time series and whiskers in inset) indicate the minimum-to-maximum range across BLUE estimates obtained by scaling BLUE carbon densities with individual DGVMs (dots indicate individual estimates; see Methods) and one standard deviation for the TRENDY and GCB2022 estimates. Our BLUE estimates are based on simulations using averaged DGVM carbon densities (see Methods); thus, they do not correspond to the mean of the BLUE estimates based on individual DGVMs. Source data are provided as a Source Data file.

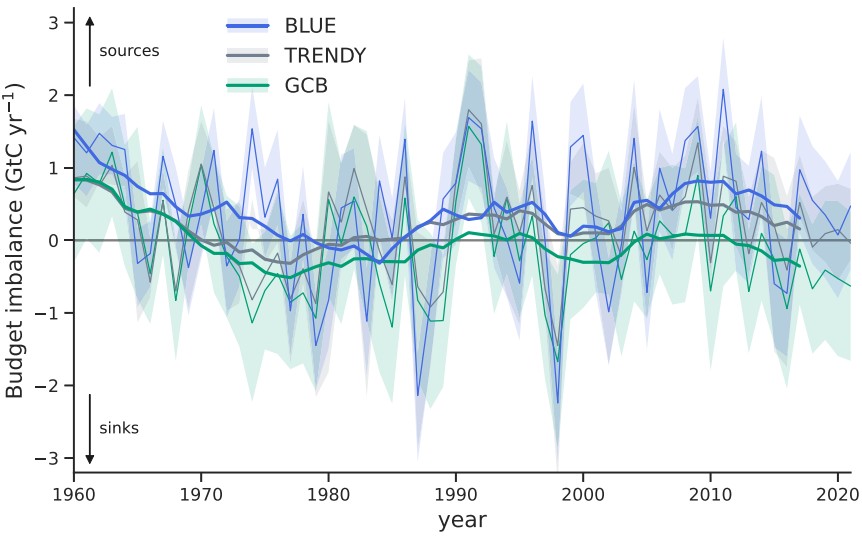

**Fig. 5 | Imbalance of the global carbon budget using the terrestrial components from BLUE, TRENDY, and the GCB.** The budget imbalance ($B_{IM}$) is a measure of the mismatch between the estimated $CO_2$ emissions from land-use change ($E_{LUC}$) and fossil fuels ($E_{FOS}$) and the estimated $CO_2$ sinks on land ($S_{LAND}$), in the ocean ($S_{OCEAN}$), and in the atmosphere ($G_{ATM}$). The bookkeeping model BLUE uses $E_{LUC,trans}$ (based on transient environmental conditions) and $S_{LAND,trans}$ (under actual, transient land cover). For TRENDY we use estimates of the net land flux from Dynamic Global Vegetation Models (DGVMs) under the TRENDY S3 simulation, which is consistent with our BLUE estimate as it uses transient environmental conditions and transient land-use change. For the Global Carbon Budget (GCB) we follow ref. 51 and use $S_{LAND}$ from TRENDY under pre-industrial land cover (corresponding to $S_{LAND,pi}$) and $E_{LUC}$ from three bookkeeping models from GCB2022 (corresponding to $E_{LUC,pd}$, see Methods for further details). For calculating the $B_{IM}$, emissions from peat fires and peat drainage from external datasets are added to $E_{LUC}$ and to the net land flux estimates (see Methods). Shading denotes uncertainties, shown as minimum-to-maximum range across BLUE estimates obtained by scaling BLUE carbon densities with individual DGVMs, as one standard deviation across DGVM estimates (for TRENDY), and combined across DGVM and bookkeeping estimates (for the GCB). Source data are provided as a Source Data file.

would be consistent with the improved $E_{LUC}$ and $S_{LAND}$ estimates proposed by our study. We thus expect that improvements in other parts of the carbon budget would bring the budget imbalance closer to zero again.

With our approach we are able to quantify all major components of the terrestrial carbon budget with BLUE: $E_{LUC}$, $S_{LAND}$, the environmental contribution to $E_{LUC}$, RSS, and LASC (as the sum of the latter two). This makes it possible to mimic other estimates of terrestrial carbon budget terms. For instance, the sum of $E_{LUC,trans}$ and RSS conceptually equals $E_{LUC,TRENDY}$ (see Methods), both showing an upward trend from the 1960s onwards due to the accumulating nature of LASC (Supplementary Fig. 10a). Similarly, we can deduce a BLUE

estimate for $S_{LAND}$ under pre-industrial land cover ($S_{LAND,pi}$) that conceptually agrees with $S_{LAND,TRENDY}$ (Supplementary Fig. 10b). The remaining gap between the conceptually similar estimates of BLUE and TRENDY (0.7 GtC yr$^{-1}$ for $E_{LUC,trans}$ + RSS vs. $E_{LUC,TRENDY}$, and 0.6 GtC yr$^{-1}$ for $S_{LAND,pi}$ vs. $S_{LAND,TRENDY}$) is due to several reasons. Those include model-intrinsic differences in the distribution of natural vegetation types, the (degree of) implementation of land management practices, and carbon densities of different types of vegetation. Particularly the latter plays an important role: Replacing the transient BLUE carbon densities with transient DGVM carbon densities in our simulations decreases $E_{LUC,trans}$ by 0.4 GtC yr$^{-1}$ and $S_{LAND,trans}$ by 1.3 GtC yr$^{-1}$ in 2012–2021 (Supplementary Fig. 10), as the carbon densities in BLUE are substantially higher than the TRENDY average for most types of vegetation (Supplementary Fig. 8). While the model spread is notoriously large for carbon densities, BLUE has been shown to be closer to observational evidence on global scale than most process-based models[7]. Globally, model-intrinsic differences and the impact of land cover on the carbon sink largely offset each other, resulting in similar estimates for $E_{LUC}$ and $S_{LAND}$ by our approach and by TRENDY models (Table 2).

Despite delivering a conceptually consistent carbon budget, our approach contains several sources of uncertainty, mainly stemming from the limited reliability of the DGVM carbon densities, our processing of carbon densities as global averages, and the LULUCF data. First, DGVMs show a large spread in their sensitivity of carbon fluxes to environmental changes, e.g., due to simplified representations of biogeochemical processes and differences in assumptions on plant productivity, plant allocation, nutrient availability, and carbon turnover times[26,30-34]. Since we scale the BLUE carbon densities with the DGVM carbon density ratios, these uncertainties of DGVMs propagate to our $E_{LUC}$ and $S_{LAND}$ estimates. The resulting uncertainties are relatively small for $E_{LUC,trans}$ (Fig. 1, Table 2). This is explained by the fact that the scaling with DGVM carbon density ratios affects both carbon emissions and removals (Fig. 1b), which partly cancel out. In contrast, the uncertainties are much more pronounced for $S_{LAND,trans}$ (Fig. 2). The reasons for this are the lack of a compensating emissions/ removals effect (as for $E_{LUC,trans}$) and that impacts of environmental changes on land areas act more homogeneously and widespread compared to LULUCF impacts (compare Figs. 1a and 2, and Figs. 1c and 3a). These uncertainties may be reduced in the future, as our DGVM-based carbon density dataset can easily be updated to new and improved versions of DGVMs or other model- or observation-based transient carbon density estimates. Second, we broadly aggregate carbon densities of different types of vegetation at the global level (see Methods). Local effects (e.g., of fire or drought if represented in DGVMs[17]), latitudinal differences in the effects of $CO_2$ and temperature on the carbon sink[35,36], and natural climate variability may thus be underrepresented in our estimates. This could explain distinct regional discrepancies between $S_{LAND,TRENDY}$ and $S_{LAND,pi}$, particularly in forested regions (Supplementary Fig. 11). Third, potential errors in the LULUCF data need to be considered[37-39], as they likely contribute to the supposed regional hotspots of emissions (Fig. 1c, Supplementary Figs. 2 and 7). For example, ref. 37 found a bias between observed biomass estimates and those simulated by BLUE in south Asia, Southeast Asia, and Equatorial Africa and attributed this bias to an overestimation of prescribed wood harvest and clearing rates in the LULUCF data. This impacts $E_{LUC}$, as generally high estimates in those regions increase more (in absolute terms) when transient environmental conditions are considered. Further, ref. 37 estimated an $E_{LUC}$ increase by 1.4 GtC yr$^{-1}$ in 2000–2019 by integrating an observation-based time series of woody vegetation carbon densities into BLUE compared to a simulation with static environmental conditions of 2000. This increase is substantially larger than our corresponding estimate of 0.2 (0.1, 0.3) GtC yr$^{-1}$ in 2000–2019. However, a direct comparison to our approach is difficult, since ref. 37 assimilates the

absolute (observed) carbon densities in BLUE, whereas we use the trends from the DGVMs to scale the BLUE carbon densities. Consequently, discrepancies between the forcing dataset LUH2 and the observed carbon density dataset used in ref. 37 may have a larger impact on $E_{LUC,trans}$ compared to our approach.

## Discussion

Typically, the two components of the terrestrial carbon budget—$E_{LUC}$ and $S_{LAND}$—are estimated without considering impacts of environmental changes on the former and of land-use changes on the latter. We have resolved the resulting conceptual inconsistency in the terrestrial carbon budget by integrating transient environmental conditions into the bookkeeping model BLUE. Our study suggests that effects of environmental changes, which are not considered in current carbon budgets, cause a 14% (8%, 23%) increase in $E_{LUC}$ globally over 2012–2021. This is crucial for correctly estimating the remaining carbon budget to limit global warming to 1.5 °C or 2.0 °C[1,40]. The same applies to the Global Stocktake, which relies on an accurate assessment of the success of climate mitigation through LULUCF by halting deforestation or by implementing re/afforestation projects. As future environmental conditions are expected to diverge further from the present-day state, $E_{LUC}$ estimates are projected to grow further apart in future decades[9], yet decisively depending on the future evolution of $CO_2$ concentrations and climate change[41].

Our $S_{LAND}$ estimate differs from previous estimates as we account for historical LULUCF, which caused a loss of valuable ecosystems, in particular forests, that would sequester much additional carbon if they still existed. Our results unveil that, by using pre-industrial land cover, standard budgeting approaches overestimate $S_{LAND}$ by 23% (8%, 33%) averaged over 2012–2021. With the presented approach, land-use effects on $S_{LAND}$ can be separated from detrimental environmental impacts, both of which can put natural ecosystems under extensive stress[42]. Losses in $S_{LAND}$ due to LULUCF are expected to increase even further because halting deforestation and forest degradation is targeted to be achieved only by 2030[43], thus further augmenting the lost sinks (i.e., RSS), which accumulate over time[16] (Table 2). Additionally, damages from climate change, such as land drying, droughts, and wildfires, may increasingly counteract the beneficial effects of increased atmospheric $CO_2$ levels on plant growth[32,44,45]. The long-term evolution of $S_{LAND}$ and RSS thus crucially depends on our climate mitigation efforts and on which of the vastly different potential future land-use paths we follow[46].

Correcting $S_{LAND}$ for the already replaced sinks/sources and considering environmental effects on $E_{LUC}$ with our spatially explicit bookkeeping model BLUE enables a greater consistency with atmospheric inversions (e.g., currently, inversions and conventional bookkeeping estimates largely differ in their estimates of carbon sinks in eastern Europe[47]) and other monitoring or modeling opportunities recently proposed for a complete reporting of $CO_2$ fluxes on managed and unmanaged land[48]. It will also contribute to further harmonizing and reducing the gap between $E_{LUC}$ estimates from global carbon cycle models and national greenhouse gas inventories by avoiding the current misattribution of replaced sinks (RSS) to non-forest natural sinks[42]. The strikingly large additional sinks that we show as already lost through LULUCF (Table 2, Fig. 2) highlight the immense value of natural ecosystems for climate regulation, thus providing an additional incentive to protect remaining forests. A consistent carbon budgeting unravels a double benefit of restoring degraded ecosystems: avoided $CO_2$ emissions and increased $CO_2$ removals through LULUCF as well as provision of additional sinks in response to environmental changes (e.g. in re/afforested regions). While our study argues that these latter sinks should be attributed to natural terms, including them under anthropogenic activities may be advantageous from a political viewpoint: counting the additional, natural sinks as activities in the LULUCF sector and thus allowing for $CO_2$ credits may incentivize carbon

dioxide removal, such as through re/afforestation. However, the future evolution of these additional sinks—as the natural land sink in general—is highly dependent on the socioeconomic pathway and mitigation efforts humanity will follow[4,41]. Land sinks may even turn to large-scale sources under future weather extremes[32,49] or when atmospheric $CO_2$ concentrations eventually decline. Such shifts in dynamics make it even more crucial to determine environmental effects separately from anthropogenic land use.

We have argued here that an intuitive accounting of land-use emissions, in line with actual observations, has to include the effects of environmental changes on the carbon stocks existing at the time of the LULUCF event (e.g., clearing, wood harvesting, re/afforestation). Additionally, estimates of the natural land sink must reflect the impacts of LULUCF activities on environmental effects by considering the transient land cover instead of a theoretical pre-industrial state. Our framework provides the tool to quantify all relevant fluxes of the terrestrial carbon budget separately and in a spatially explicit way. This not only delivers a fully consistent terrestrial carbon budget on its own, but would also make it possible to correct the current GCB estimates by subtracting RSS from $S_{LAND}$ and by replacing $E_{LUC,pd}$ by $E_{LUC,trans}$. Further, it is crucial to link reporting and certification frameworks, the development of which are currently widely underway[50], directly with the scientific carbon budget approach and thus to avert double-counting of $CO_2$ sinks or omission of sources, which may otherwise put reaching national and global climate targets at risk.

## Methods

### Data

**TRENDY data.** For our study we use data from the S2 and S3 simulations of Dynamic Global Vegetation Models (DGVMs) conducted for the Global Carbon Budget 2022[51] (TRENDYv11). The S2 simulation accounts for transient environmental conditions of climate, atmospheric $CO_2$ concentration, and nitrogen deposition but keeps land cover at its pre-industrial state (i.e., it excludes land-use changes after 1700). The S3 simulation uses the same environmental forcing as S2 and additionally includes transient land-use changes, employing land-use change data from the Land Use Harmonization 2 dataset for the GCB2022 (LUH2-GCB2022, ref. [52] updated according to ref. [51]). Further information on input datasets for the respective TRENDY simulation setups can be found in ref. [51]. The natural land sink ($S_{LAND}$) is calculated for 15 DGVMs (excluding IBIS) following the approach used by the GCB2022, i.e., using the annual global sum of the variable 'net biome productivity' (NBP) of the S2 simulation ($S_{LAND,TRENDY}$). The DGVM IBIS was excluded from further analysis as its $S_{LAND}$ estimate indicated in the GCB supplementary could not be reproduced with the TRENDY data. All DGVMs considered in GCB2022 provide data for NBP, whereas only few DGVMs provide output specific to plant functional types (PFTs) for vegetation and soil carbon, which we use for the scaling of the default BLUE carbon densities: We use PFT-specific annual carbon densities for vegetation (variable cVegpft) and soil (cSoilpft), and the corresponding land-cover fractions (land-CoverFrac) from the TRENDY S2 simulation. We process data from eight DGVMs for vegetation (CABLE-POP, CLASSIC, JSBACH, JULES, LPJ-GUESS, ORCHIDEE, SDGVM, and YIBs) and from five DGVMs for soil (CABLE-POP, CLASSIC, JSBACH, ORCHIDEE, and YIBs), as only those DGVMs provide the required PFT-specific output. For JSBACH we use data from the TRENDYv12 version (which is used in the GCB2023), as the PFT-specific data submitted to TRENDYv11 could not be used due to an error in the simulation setup. To identify the time-variant fractions of crop and pasture, we use the land-cover fractions of crop and pasture from the S3 simulation. To be consistent with the GCB2022 $S_{LAND}$ estimate, we calculate $S_{LAND,TRENDY}$ using all DGVMs considered in the GCB2022 (except IBIS), i.e. 15 DGVMs. Note that the differences in $S_{LAND,TRENDY}$ based on all 15 DGVMs and based on the subset of eight DGVMs are only minor (Supplementary Fig. 12). The net land flux is

estimated for all 15 DGVMs using NBP from the S3 simulation. The difference between the NBP estimates of the S3 and S2 simulations is used as the estimate of land-use change emissions ($E_{LUC}$) for TRENDY (see also equations further below).

**Other data.** Further data that we use are fossil $CO_2$ emissions ($E_{FOS}$), the atmospheric $CO_2$ growth rate ($G_{ATM}$), and the ocean $CO_2$ sink ($S_{OCEAN}$), all taken from the GCB2022[51]. We also use the $E_{LUC}$ estimates from GCB2022, which stem from the three bookkeeping models H&C2023 (denoted 'updated H&N2017' in GCB2022), OSCAR, and the present-day simulation of BLUE (see further below regarding the different BLUE simulations used here). We add emissions from peat fires and peat drainage from GCB2022 to estimates of the net land flux for BLUE, TRENDY, and the GCB (see ref. [51] for details about the derivation of peat emissions). The net land flux of the GCB is the sum of $S_{LAND}$ from TRENDY under pre-industrial land cover and $E_{LUC}$ from the three bookkeeping models. We further use estimates of the net land flux based on atmospheric $O_2$ observations and based on atmospheric inversion systems from GCB2023[1]. GCB2023 indicates these estimates for the period 2013–2022. To compare with the period used in our study (2012–2021), we employed data for eight atmospheric inversion systems that provide data for the period 2012–2021[53]. We also obtained an additional estimate of the net land flux from atmospheric $O_2$ observations for 2010–2019 (M. O'Sullivan, personal communication 11.04.2024), which amounts to $-1.2 \pm 0.8$ GtC yr$^{-1}$, and is thus the same as the 2013-2022 estimate indicated in Table 2.

### Uncertainty estimation

Our BLUE estimates for the different terms of the terrestrial carbon budget are based on a simulation that combines carbon density data from eight DGVMs for vegetation and from five DGVMs for soil (see Methods section above on TRENDY data). We additionally perform five individual BLUE simulations using carbon density data from the five individual DGVMs that provide carbon density data for both vegetation and soil (CABLE-POP, CLASSIC, JSBACH, ORCHIDEE, and YIBs). The minimum and maximum values of these five simulations are used as uncertainty bounds for the BLUE estimates. Note that the average of these five simulations does not correspond to our BLUE estimates (the latter are based on separate simulations using averaged DGVM carbon densities) due to non-linearities in the equations implemented in BLUE and due to the consideration of three additional DGVMs for vegetation carbon densities in the simulation that provides our BLUE estimates. $E_{LUC,pd}$ does not have an uncertainty estimate, as we require that all DGVM-scaled carbon densities match the standard BLUE carbon density under present-day conditions, and thus all simulations would yield the same estimate for $E_{LUC,pd}$.

Uncertainties for other data (TRENDY, GCB[51], atmospheric $O_2$ observations[1], data from ref. [16], data from ref. [9]) are indicated as one standard deviation around the mean. Uncertainties for the GCB estimate of net land flux and for the GCB estimate of the budget imbalance are derived by propagating $E_{LUC}$ and $S_{LAND}$ uncertainties. For the atmospheric inversion estimate, the uncertainty is indicated as minimum-to-maximum range across eight inversions.

### Implementation of transient environmental forcing into BLUE

**Model description of the Bookkeeping of Land Use Emissions model (BLUE).** The Bookkeeping of Land Use Emissions model BLUE[7] is a spatially explicit semi-empirical bookkeeping model that simulates carbon fluxes from LULUCF by tracking the carbon content in atmosphere, vegetation, soil components (undergoing fast and slow relaxation processes), and harvested wood product pools (filled after wood harvest and forest clearing, with products decomposing on time scales of 1, 10, or 100 years) for four land-cover types (primary land, secondary land, cropland, and pasture) and eleven natural PFTs. All carbon pools are initialized with an equilibrium carbon content per

area ρ, i.e., the carbon densities. ρ is defined individually for each land-cover type on PFTs 1–11[7]. Prior to LULUCF activities, all carbon pools are in equilibrium. If a LULUCF activity occurs, the carbon pools are transferred from their equilibrium state into a disequilibrium state. BLUE distinguishes between the following LULUCF activities: abandonment (land-cover change from crop or pasture to secondary land), clearing for cropland or pasture (land-cover change from primary land or secondary land to crop or pasture), wood harvest (land-cover change from primary to secondary land or land management on secondary land), and transitions between crop and pasture. Upon a land-use transition, carbon is transferred from one land-cover type (source cover type, $j$) to another (target cover type, $j'$). The amount of carbon removed from $j$ and added to $j'$ depends on ρ and on the area affected by the land-use transition. Carbon is removed from the vegetation or soil equilibrium carbon pool of $j$ (and from the excess carbon pool of $j$ in case it is not zero) and is distributed to the excess pools of soil (slow and fast pools) and products in $j'$. Additionally, a new equilibrium pool is defined with the equilibrium carbon content of $j'$. The amount of carbon that $j'$ would reach in equilibrium is subtracted from (or added to) the excess pool of $j'$, as this is the amount of carbon missing or in excess to reach the new equilibrium. In each timestep, the excess carbon pools change according to the respective relaxation time constants (which follow exponential functions) defined for each cover type, each PFT, and each carbon pool. The temporal evolution of the carbon content of each carbon pool in a disequilibrium state (i.e., after a land-use transition) is determined by the new equilibrium carbon content it is supposed to reach (i.e., the equilibrium carbon content of the new land-use state $j'$), by the amount of carbon that is missing or in excess to reach that equilibrium carbon content, and by the relaxation time of each process. Eventually, carbon is released to the equilibrium pool of the atmosphere. Each simulation with BLUE provides annual output of the sum of the equilibrium and excess pools for each carbon pool. The annual change of the carbon content in the atmospheric pool ($\Delta C_A$) corresponds to $E_{LUC}$. A complete description of the BLUE model can be found in the supplementary information of ref. 7.

**Integrating transient environmental effects in BLUE.** The original purpose of a bookkeeping model is to isolate effects of anthropogenic drivers, unimpaired by environmental changes. For this study, we advance the bookkeeping model BLUE such that (1) effects of transient environmental conditions on anthropogenic carbon fluxes are captured and (2) the effects of environmental changes can be isolated from those of land-use changes. (1) delivers an estimate of $E_{LUC}$ that would realistically be released into the atmosphere upon a LULUCF activity as it accounts for all transient processes, such as effects of rising $CO_2$ levels, climate change, and nitrogen deposition. Furthermore, our developments make it possible to use different sets of carbon densities in BLUE based on the specific study needs to estimate $E_{LUC}$ under transient, present-day (as in the default setup of BLUE), and pre-industrial conditions. (2) allows us to quantify $S_{LAND}$. In contrast to the TRENDY S2 simulation, which estimates $S_{LAND}$ under pre-industrial land cover, we quantify $S_{LAND}$ under the actual, transient land cover. In this way we can exclude the hypothetical $CO_2$ sinks lost through ecosystem degradation (i.e., the replaced sources and sinks[15], RSS; see Table 1), which are included if assuming a pre-industrial land cover. We define $S_{LAND}$ as the environmental effects on carbon pools on natural land (i.e., land that has never been impacted by LULUCF activities or has fully recovered from LULUCF activities) and managed land (land disturbed by direct LULUCF activities due to recent or ongoing LULUCF or land that is still recovering from past LULUCF activities[14]). Note that we assign the effects of transient environmental conditions on anthropogenic carbon fluxes to $E_{LUC}$ until the carbon stock prevalent at the time of the LULUCF event is reached again. The subsequent environmental effects due to ongoing environmental changes are assigned to $S_{LAND}$. We derive $S_{LAND}$ by first subtracting annual

carbon stock changes of two BLUE simulations, one excluding and one including environmental changes, and subsequently subtracting the environmental effects on $E_{LUC}$ (see below for details). Environmental effects are included in BLUE based on transient carbon densities from DGVMs, as detailed in the next section.

### Preparation of transient carbon densities

Global observation-based time series of carbon densities are not available from the pre-industrial era onwards. We thus use the temporal evolution of carbon densities from DGVMs in the TRENDY S2 simulation to scale the time-invariant carbon densities used as default by BLUE. The scaled BLUE carbon densities thus reflect the transiently changing environmental conditions as simulated by the DGVMs.

The processing includes the following steps:

(1) The DGVM PFTs are translated to the BLUE PFTs (different types of forests, shrubland, grasses, and tundra) for the BLUE land-cover types primary land, secondary land, pasture, and cropland (see Supplementary Data 1 and 2). As the number and definition of PFTs is not standardized in DGVMs, not all PFTs can be directly matched to the BLUE PFTs. In some cases, a spatial mask is applied to PFTs of some DGVMs to fit the spatial extent of the BLUE PFTs (see Supplementary Fig. 13). In other cases, particularly for shrub PFTs, the carbon density of a BLUE PFT is constructed by weighting different PFTs of a DGVM following the cross-walking table by ref. 54 (see their Table 2). A detailed description of the mapping from DGVM PFTs to BLUE PFTs can be found in the Supplementary Methods.

(2) Annual global averages of carbon densities are calculated for each mapped PFT and for every DGVM, weighting the carbon densities by the land-cover fraction of the respective PFT in each grid cell (to reflect that grid cells with large fractions contribute more to the estimated global carbon density than grid cells with small PFT fractions). The resulting time series of global carbon densities are smoothed with a 20-year moving average. For the last nine years of the simulation period (for which the moving average did not yield values), carbon densities are linearly extrapolated based on the last 20 years of unsmoothed data.

(3) For each DGVM and each PFT, we calculate the ratio of the carbon density time series relative to the carbon density in the year 1980, as the carbon densities from ref. 10 used in the standard version of BLUE are approximately representative for that year. The resulting time series of carbon density ratios are then multiplied by the default BLUE carbon densities to derive transient carbon densities. The pre-industrial carbon densities are the 1720–1740 average of the scaled carbon densities.

We further tested the sensitivity of BLUE to a different set of carbon densities. For this purpose, we produced a time-series of the absolute carbon densities from DGVMs as input for BLUE to compare with the scaled BLUE carbon densities. Note that also in this sensitivity test we kept the ability of BLUE to account for degradation from primary to secondary land, which is usually lacking in DGVMs. To derive the carbon densities of secondary land in the sensitivity test, we multiply the DGVM carbon densities for primary land by the fraction of the secondary to primary carbon densities in BLUE.

### BLUE simulations

We employ five different simulation setups in BLUE: (1) $BLUE_{pi}$, (2) $BLUE_{pd}$, (3) $BLUE_{trans}$, (4) $BLUE_{trans+m}$, and (5) $BLUE_{S2}$. We employ the LUH2-GCB2022 dataset as land-use change data. Our BLUE output thus has the same spatial resolution as LUH2-GCB2022, namely 0.25°. The simulation setups (1)-(3) apply different environmental conditions and are used to derive three different definitions of $E_{LUC}$ while (1), (3) and (4) are used to estimate $S_{LAND}$. The first simulation ($BLUE_{pi}$) applies time-invariant pre-industrial carbon densities for the whole simulation period. The second simulation ($BLUE_{pd}$) corresponds to the default

BLUE set-up and applies time-invariant carbon densities that approximately represent present-day carbon densities (i.e., of the early 1980s[7]). The third (BLUE$_{trans}$) and fourth (BLUE$_{trans+m}$) simulations use the transient carbon densities derived from DGVMs, as described above. Transient carbon densities are used from 1730 onwards, as the carbon densities in 1730 correspond to the 1720–1740 average used for model initialization. Note that in the pre-industrial and transient simulations, equilibrium carbon pools are initialized with pre-industrial carbon densities, while in the default simulation (BLUE$_{pd}$) present-day carbon densities are used for the initialization. BLUE$_{trans}$ accounts for environmental effects previous to a LULUCF activity (by using the transient carbon densities). BLUE$_{trans+m}$ additionally includes environmental effects following a LULUCF activity, accounting for environmental effects on natural and managed land altering the carbon stocks over time (e.g., forests typically grow denser, as suggested by the DGVM carbon densities, Supplementary Fig. 3). For this purpose, the equilibrium and excess pools of vegetation and slow soil are multiplied by the fraction of the carbon density ratios of one year to the previous year in each time step, for each PFT, and for each land-cover type (note that we assume that the fraction of carbon going to the rapid soil pools and product pools is not altered by environmental effects). Within the BLUE model structure, this leads to an increase of the missing or excess carbon that is needed to reach equilibrium. Forests that are regrowing after clearing or wood harvest thus not only grow back to their carbon density at the time of their clearing (regrowth due to recovery from past management, as captured in BLUE$_{trans}$) but grow even denser over time due to more favorable environmental conditions (regrowth and further growth due to environmental changes). Lastly, BLUE$_{S2}$ is similar to BLUE$_{trans+m}$ in the sense that it captures environmental effects but it excludes all land-use changes. Therefore, it conceptually resembles the TRENDY S2 simulation.

## Derivation of land-use emissions and of the natural land sink

We use several output variables from the five different BLUE simulations to derive $E_{LUC}$ and $S_{LAND}$. In the following, we employ the conceptual framework of ref. 14 to describe the carbon fluxes of each BLUE simulation. The framework distinguishes between $E$ and $L$, the prefix $\delta$, and the subscripts p, m, and n. $E$ represents the carbon fluxes induced by environmental changes. $L$ represents the carbon fluxes due to LULUCF (i.e., $E_{LUC}$). $\delta$ describes the effect of environmental changes on a variable (our setup captures this by the transient carbon densities), and p, m, and n indicate potential natural (p), managed (m), and natural (n) land. Potential natural land refers to the land cover as it would exist without human interventions.

To estimate $E_{LUC}$ under pre-industrial, present-day, and transient environmental conditions, we use the output of the annual change in the atmospheric pool ($\Delta C_A$) of the simulations in Eqs. (1)–(3):

$$E_{LUC,pi} = \triangle C_{A,BLUE_{pi}} = L \tag{1}$$

$$E_{LUC,pd} = \triangle C_{A,BLUE_{pd}} = L + \delta_{pd}L \tag{2}$$

$$E_{LUC,trans} = \triangle C_{A,BLUE_{trans}} = L + \delta L \tag{3}$$

All three simulations include carbon stock changes due to LULUCF ($L$) but vary in their consideration of environmental effects on $E_{LUC}$ ($\delta L$). $E_{LUC,pi}$ excludes all environmental effects, $E_{LUC,pd}$ includes environmental effects on $E_{LUC}$ based on present-day environmental conditions, and $E_{LUC,trans}$ includes transient environmental effects on $E_{LUC}$. Note that $\delta$ without subscript denotes $\delta_{trans}$.

To derive $S_{LAND}$ we make use of multiple simulations and use the annual carbon change of the vegetation, soil, and product pools ($\Delta C_L$). $\Delta C_L$ of the BLUE$_{S2}$ simulation resembles the S2 simulation of TRENDY

models and corresponds to $S_{LAND}$ under pre-industrial land cover as it excludes LULUCF:

$$S_{LAND,pi} = \triangle C_{L,BLUE_{S2}} = E_n + E_p + \delta\left(E_n + E_p\right) \tag{4}$$

$S_{LAND,pi}$ thus includes hypothetical land sinks, which in reality are lost due to historical ecosystem degradation. Note that $E_n$, $E_p$, and $E_m$ are considered to be zero over longer time scales as effects through natural climate variability largely cancel out, and thus only terms affected by the $\delta$ operator remain.

Transient $S_{LAND}$ as it occurs in reality (i.e., based on transient land cover) is defined as:

$$S_{LAND,trans} = \delta\left(E_n + E_m\right) \tag{5}$$

To derive the natural land sink on transient land cover, we use $\Delta C_L$ of BLUE$_{trans+m}$ and of BLUE$_{pi}$. $\Delta C_L$ in BLUE$_{trans+m}$ is caused by LULUCF and by environmental effects on $E_{LUC}$ and on carbon pools of natural and managed land. It can therefore be written as:

$$\triangle C_{L,BLUE_{trans+m}} = E_n + E_m + L + \delta\left(E_n + E_m + L\right) \tag{6}$$

Note that $\triangle C_{L,BLUE_{trans+m}}$ yields the net land flux and thus resembles the TRENDY S3 simulation. In contrast, the annual changes in the pre-industrial simulation setup ($\triangle C_{L,BLUE_{pi}}$) are only driven by LULUCF, since environmental effects are excluded:

$$\triangle C_{L,BLUE_{pi}} = E_n + E_m + L \tag{7}$$

The difference between Eq. (6) and Eq. (7) thus yields the environmental effects on $E_{LUC}$ and on the carbon fluxes on natural and managed land $\delta(E_n + E_m + L)$. To derive $S_{LAND}$ under transient land cover we need to subtract $\delta L$, which we obtain as the difference between Eq. (3) and Eq. (1):

$$S_{LAND,trans} = \left(\triangle C_{L,BLUE_{trans+m}} - \triangle C_{L,BLUE_{pi}}\right) - \left(\triangle C_{A,BLUE_{trans}} - \triangle C_{A,BLUE_{pi}}\right)$$
$$= \delta\left(E_n + E_m\right) \tag{8}$$

$S_{LAND,trans}$ is our pursued definition of $S_{LAND}$ as it yields the natural land sink under transient land cover. The difference between $S_{LAND,trans}$ and $S_{LAND,pi}$ (Eq. (5) minus Eq. (4)) yields the lost (or gained) sinks through historical degradation (or restoration) of ecosystems, denoted as replaced sinks and sources[15] (RSS):

$$RSS = S_{LAND,trans} - S_{LAND,pi} = \delta\left(E_m - E_p\right) \tag{9}$$

RSS closely relates to the loss of additional sink capacity (LASC), which is defined by ref. 16 as:

$$LASC = \delta L + RSS = \delta\left(L + E_m - E_p\right) \tag{10}$$

Using Eqs. (1)–(10), we can derive all relevant carbon budget terms with BLUE: $E_{LUC}$, $S_{LAND}$, the environmental contribution to $E_{LUC}$ ($\delta L$), RSS, and LASC.

## DGVM estimates of the terms in the terrestrial carbon budget

The GCB estimate of $S_{LAND}$ stems from the TRENDY S2 simulation, which employs transient environmental conditions but keeps land use and land cover at its pre-industrial state. This simulation yields $S_{LAND,pi}$ according to Eq. (4), thus including the hypothetical land sink, which, in reality, is lost due to historical ecosystem degradation (RSS).

The TRENDY $E_{LUC}$ estimate is calculated by subtracting $S_{LAND,pi}$ from the net land flux, which is obtained from the S3 simulation, in

which both environmental conditions and land cover are transient (yielding the net land flux according to Eq. (6)). $E_{\text{LUC}}$ from DGVMs thus reads:

$$E_{\text{LUC,DGVMs}} = L + E_{\text{m}} - E_{\text{p}} + \delta\left(L + E_{\text{m}} - E_{\text{p}}\right). \tag{11}$$

In contrast to $E_{\text{LUC,trans}}$ (Eq. (3)), $E_{\text{LUC,DGVMs}}$ thus includes RSS (Eq. (9)).

LASC can be calculated according to Eq. (10) using additional DGVM simulations[16]. With the current TRENDY simulation setup, it is however not possible to separate the transient effects on $E_{\text{LUC}}$ (i.e., $\delta L$) from the other transient effects (i.e., $\delta E_{\text{m}}$, $\delta E_{\text{p}}$). Thus, $E_{\text{LUC,trans}}$, $S_{\text{LAND,trans}}$, and RSS cannot be estimated with DGVM data.

## Data availability

The processed data are available at https://doi.org/10.6084/m9.figshare.26431516. Source data are provided with this paper.

## Code availability

The code for the analysis in this paper is available upon request to the corresponding author.

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

## Acknowledgements

We thank Ingrid Luijkx for providing atmospheric inversion estimates for 2012–2021 and Mike O'Sullivan for providing data on atmospheric $O_2$ observations for 2010-2019. This work was supported by the German Federal Ministry for Research and Education (BMBF) through the research project STEPSEC, Grant Number: 01LS2102A. This work used resources of the Deutsches Klimarechenzentrum (DKRZ) granted by its Scientific Steering Committee (WLA) under Project ID bm0891. S.B. acknowledges support from the German Stifterverband für die Deutsche Wissenschaft e.V. in collaboration with Volkswagen AG (Addressing the new role of terrestrial $CO_2$ fluxes for climate mitigation).

## Author contributions

C.S., J.P. and L.D. designed the study. L.D. conducted the integration of transient carbon densities into the model and performed the simulations with input from S.B. L.D. analyzed the data with input from all authors. L.D. drafted the initial version of the manuscript, and all authors provided critical feedback and helped shape the research, analysis and manuscript. All authors contributed to the review and editing of the manuscript.

## Funding

## Competing interests

The authors declare no competing interests.
