## [Peer Review File · Nature Communications]

A consistent budgeting of terrestrial carbon fluxesREVIEWER COMMENTS

Reviewer #1 (Remarks to the Author):

This study was initiated from a very interesting point and aimed to resolve the discrepancies in the quantification of ELUC and ELAND, which resulted from different model approaches and data sources. However, after reading this manuscript, three concerns come to my mind.

Firstly, as the authors stated, they used transient estimates from the DGVMs as input for their bookkeeping BLUE models. This approach likely introduces errors or uncertainties from the DGVM model, thereby potentially doubling or amplifying these errors and uncertainties, which could render the results unreliable. Furthermore, the subsequent comparison between ELUC and ELAND appears somewhat circular, even though there are no better solutions available.

Secondly, given the existence of process-based DGVMs, I wonder why a similar model for ELUC wasn't constructed, which could potentially offer improvements over the bookkeeping BLUE approach. The latter still partially suffers from the issues mentioned in the context of older bookkeeping models.

Thirdly, as the authors mentioned in lines 108-113, the significant difference between the estimates from the observed time-series data-based model and the DGVM-derived transient data-based model indicates that the approach used may not accurately represent reality. The discussion of bias in lines 200-217 attempts to identify potential sources of uncertainty. However, validating the DGVM-based estimates with actual observed data could lend more credibility to this study.

The paper 'Modelled land use and land cover change emissions – a spatio-temporal comparison of different approaches' by the last author emphasizes an approach that provides a more robust accounting of fLULCC, for example, by estimating a mean DGVM ensemble fLULCC and LASC for a defined reference period and under homogeneous environmental changes (CO₂ only). It would be interesting to understand the differences between the approach used in these two studies. If they are not fundamentally different, it could question the novelty of this study."

DGVMs are very important and informative for understanding the carbon cycles in global ecosystems; however, to accurately benchmark the global carbon budget, we should integrate more real observed data into these models.

Reviewer #2 (Remarks to the Author):

The inconsistency of CO₂ emissions from land-use change (ELUC) and of the terrestrial CO₂ sink significantly affects evaluation of carbon neutrality. Dorgeist et al. developed a new model named BLUE to precisely quantify the budgeting of terrestrial carbon fluxes. Generally, the topic is

important and the scientific question is meaningful. The paper structure and the writing is good. However, based on my personal evaluation, I have several concerns for the BLUE model and the results.

Main comments

#1 Is the Imbalance of the global carbon budget in the BLUE affects the carbon evaluation results?

To my personal understanding, Extended Data Fig. 1 showing that the imbalance of the global carbon budget in the BLUE is much higher than GCB and TRENDY. If the model is robust, the imbalance of BLUE should be around 0. However, the BLUE model seldom reach zero so an imbalance of BLUE may affect the terrestrial carbon fluxes evaluation. The carbon imbalance in BLUE may reach 1 Gt yr⁻¹ around 2005 to 2020. Compared with Fig.1 in the main text, the carbon flux from terrestrial ecosystems is only 0.3 Gt yr⁻¹, which is much less than the imbalance of BLUE (i.e. 1 Gt yr⁻¹). Therefore, whether the imbalance of BLUE affect the results, I have no idea about this.

#2 Is there any ground surveying data or independent data source of carbon fluxes can prove that the BLUE model is robust?

This study only compared the data from simulation (i.e. GCB2022, TRENDY, BLUE, OSCAR). The critical question is all of these data are just from 'model world' but no evidence showing this model can reflect the 'real world'. One possible solution is the authors may apply the ground surveying data such as carbon fluxes site data or the NEP data in FLUXCOM data to show whether the BLUE model can reflect the real world.

#3 The figures are nice but sometimes hard to read.

As this study is quantifying the carbon sink and source, I suggest all the figures can add a sign for whether the carbon fluxes represent for a carbon sink or source. For example, the Fig.1C, Fig.2, Fig.4 can add a sign for whether it is a carbon source or sink for positive and negative value of carbon fluxes.

Specific comments

L7 'We find that state-of-the-art process-based models overestimate SLAND by 23% in 2012-2021 as they include hypothetical sinks that in reality are lost through historic ecosystem degradation.' So the overestimation of SLAND will lead to what kind of results to the carbon neutrality? The authors may revise this sentence.

L9 'Additionally, ELUC increases by 14% in 2012-2021 when considering environmental effects.' This also need to be revised. The increases of ELUC will delay or accelerate the carbon neutrality? Fig.1 One of the critical question is, why the author showing the BLUE output only from 2012 to 2021 in subplot b and c? Since the BLUE model can be extended from 1850 to present. Are these study years unique? Will it affect the carbon fluxes evaluation? For example, if the study years are from 1980 to 2021 or 2001 to 2021, will the CO₂ fluxes being quite different (Fig.1b)? For more results, the authors may consider adding a new figure or table in the SI.

Table1. This is another main concern for me. The BLUE model can express the ELUC, SLAND, Net land flux from 1850 to 2021. But why the comparison of BLUE to the existed model or results are not consist? The BLUE output can also match the years (2009-2018) in Obermeier et al. with just adding

a new line for their results. So one solution is the authors can focus on a certain time range (2012-2021) and highlight (one or two sentences at the introduction) whether they just compared the carbon fluxes at these years.

Fig.4 When compared this results with Extended Fig.1, I am doubting about whether the imbalance will significantly affect the output of BLUE. If the imbalance or called as the uncertainties (Extended Fig.1) adding in Fig.4, what will the BLUE series results look like?

As I cannot evaluate whether the model is robust, I cannot give another more comments for the specific number or conclusions showing in the main text.

Here are also some suggestion for improving the methods.

L289 why choosing 8 models from TRENDY and called them as TRENDY output? The GCB2022 (TRENDY v11) also content these 8 models. The question is, why separated the TRENDY v11 into two results (GCB, TRENDY)?

L366 The improvement of RSS is important. Can the authors prove a full paragraph or a new figure on it? I also suggest the authors adding a concept figure for what is input and output of BLUE model and what is the improvement of BLUE to overcome the previous shortcomings. So a concept figure at the SI will be an alternative.

Reviewer #3 (Remarks to the Author):

Overview

The authors update a land use change bookkeeping model with transient environmental conditions from the TRENDY ensemble. In the Global Carbon Budget, land use change emissions and the natural carbon sink are estimated independently and therefore are incompatible. Through incorporating the effects of climate change on land use change and vice-versa, the authors claim that the terrestrial carbon sink is largely overestimated. They suggest that the GCB does not consider historic ecosystem degradation or the effect of environmental changes (like CO₂ fertilization) that increase the land sink and thereby increase CO₂ emissions under deforestation. This is a challenging effort, and I congratulate the authors on undertaking this work.

In general, this paper may be an interesting step forward in bridging separate modeling approaches of the terrestrial carbon sink and anthropogenic interventions. However, I have several major concerns that the authors must address before publication.

Major comments

- I find the magnitude of the difference between TRENDY and the BLUE output to be quite astonishing (-9 GtC from GCB, vs. +53 GtC in current study) (line 167). These values do not align with estimates of the land carbon sink from atmospheric carbon isotopes, atmospheric inversions, or land sink estimates from biomass inventories (see Fig 1. Ruehr et al. 2023 Nature Reviews Earth

& Environment). The authors do not thoroughly contextualize this result within the literature. Only two references are used in Table 1 to discuss the cumulative net land flux 1850-2018, which suggest widely different values of opposite sign than the authors report (although with large certainty ranges), and little explanation is given for large this divergence. Additional discussion and observational evidence in support of their claim is necessary.

- In several places in the text, I found the writing unnecessarily confusing, with long sentences and lots of parenthetical comments. See line comments below for details. The introduction, results and discussion should be edited to prioritize clarity. I recommend breaking apart long sentences into multiple sentences and avoid unnecessary complexity.

- Consider changing the title. The current title is not related to your claim, which is that the terrestrial biosphere has been a net source of carbon since 1850 after accounting for dynamic environmental effects on land use change and vice-versa. The current title does not tell us much about your conclusions or why this paper is interesting.

- Consistent labeling and providing definitions would benefit clarity. A table of definitions is needed -- LULUCF, LASC, RSS, Sland, Eluc, natural vs. net land sink, etc. Is Fig 4 Y axis ('net land flux') the same as Fig 3 shading of Fig 2 Y axis ('CO2 flux')? Why different units for panel C of Fig 1, if this too could be CO2 flux?

- I suggest additional focus on Eluc,trans in helping to clarify the text. Having pre-industrial, present-day, and transient conditions is interesting, but it is also more confusing. Consider instead focus on the difference between Eluc,pi or Eluc,pd and ELUC,trans. You might also consider taking differences between ELUC,pi or ELUC,pd and ELUC,trans in Figs S4 and S5 to better visualize the difference over time, since it is so close in magnitude and hard to see.

Line comments

- Abstract: in your implications, add that the carbon sink is weaker (and perhaps a source since 1850) than previously thought. This overestimate of the land carbon sink may undermine efforts to limit CO2 pollution.

- Lines 19-21: unclear wording. Do "changes" refer to trends in LULUCF and environment, or are these just average fluxes over the period listed?

- Line 24: reword "separately indicate" to "uses separate methods to estimate"

- Line 40-44: split into multiple sentences, this is too long and confusing

- Line 50: Distinction between RSS and Eluc is confusing. Please provide further clarity or table of definitions.

- Line 79 onward: Provide information on what these changes are relative to

- Line 85: "This may..." Provide more implications. Over long time periods (and define scale of these), what might we expect the effect of environmental changes on Eluc to be?

- Lines 79-92: This paragraph is very clear, primarily because it provides concrete examples. I would suggest moving this paragraph up to beginning of results, or providing some of these examples within your introduction to better set the stage.

- Line 88: replace "met" with "occurred within"

- Line 131: Break into two sentences. Put parentheses into a separate sentence.

- Line 175: I am confused about the RSS and Eluc not being able to be separated. More information here would be useful.

- Lines 188-195: reported changes are relative to the ELUC without climate effects, correct? Spell

that out more clearly. Consider creating a figure focusing on this difference (ELCUC_pd vs. ELUC_trans) as this seems to be closer to your main conclusion.

- Move lines 230-233 to introduction. This is a good explanation in plain English that would benefit lay readers earlier in the text.

- Similarly, lines 451-453 are a helpful definition that could be moved up to the intro or results sections.

Figure comments

- Fig 1: interesting that ELUC_pd and ELUC_trans are so similar until ~2000. Why?

- Fig 2: Unclear what black arrow refers to. Difference between BLUE and TRENDY? Clarify in caption. Also clarify blue arrow.

- Fig S1: Mostly for my own curiosity – I see lines on country borders (between USA and Canada or PNG/Indonesia) where emissions from land-use change flip from negative to positive. Curious how country boundaries affect the model output. Perhaps include some information here. Also, why such strong but opposite trends in northern India / Bangladesh area?

- Fig S3: hard to see differences between models as they are grouped. Consider adjusting y-axes to smaller range; the data are much smaller in range than the axes.

- Fig S6: Scales on Y-axes change for each regional estimate. Consider adding an additional figure (box plot?) to more easily compare regions. You might show Sland,act cumulative for each region on one figure with same Y scale for relative differences.

Response to the reviewers' comments on manuscript "A consistent budgeting of terrestrial carbon fluxes"

Author Response to Reviewer #1:

This study was initiated from a very interesting point and aimed to resolve the discrepancies in the quantification of ELUC and SLAND, which resulted from different model approaches and data sources. However, after reading this manuscript, three concerns come to my mind.

Response: We thank the Reviewer for the constructive feedback and appreciate the valuable recommendations. We revised the manuscript accordingly as described in the following.

Firstly, as the authors stated, they used transient estimates from the DGVMs as input for their bookkeeping BLUE models. This approach likely introduces errors or uncertainties from the DGVM model, thereby potentially doubling or amplifying these errors and uncertainties, which could render the results unreliable. Furthermore, the subsequent comparison between ELUC and SLAND appears somewhat circular, even though there are no better solutions available.

Response: Thank you for pointing this out. We share the general concern that DGVMs introduce uncertainties into our estimates of E_{LUC} and S_{LAND} . As highlighted in lines 275-307, these uncertainties arise from differences in the process representation of biogeochemical processes and differences in assumptions on plant productivity, plant allocation, nutrient availability, and carbon turnover times leading to a large spread in the sensitivity of carbon fluxes to environmental changes (e.g., O'Sullivan et al. (2022), <https://doi.org/10.1038/s41467-022-32416-8>). However, uncertainties could be reduced in the future when updating our set-up with new and improved versions of DGVMs. To improve transparency on the uncertainties introduced by the DGVMs, we included an uncertainty range for our estimates (see Figs. 1, 2, 4 and Table 2), based on scaling BLUE with individual DGVMs (see lines 429-447 in the Methods section for details).

Secondly, given the existence of process based DGVMs, I wonder why a similar model for ELUC wasn't constructed, which could potentially offer improvements over the bookkeeping BLUE approach. The latter still partially suffers from the issues mentioned in the context of older bookkeeping models.

Response: There are several reasons why E_{LUC} in budget estimates is commonly derived from bookkeeping models (BMs): First, being based on empirical relationships rather than process description, BMs are more traceable and - more importantly - make it possible to include observation-based data on the carbon and ecosystem response to land-use/management changes. Second, there is a fundamental obstacle to using DGVMs directly to quantify E_{LUC} : DGVMs isolate the land-use fluxes by running a simulation with and one without land-use changes. However, this includes the "loss of additional sink capacity (LASC)" term, which is a non-realized C flux and should therefore be excluded from E_{LUC} (Obermeier et al., 2021). With our approach we solve the LASC issue while preserving the benefits of the bookkeeping approach, such as traceability to specific LULUC events and the separability of E_{LUC} into its component fluxes.

Thirdly, as the authors mentioned in lines 108-113, the significant difference between the estimates from the observed time-series data-based model and the DGVM-derived transient data-based model indicates that the approach used may not accurately represent reality. The discussion of bias in lines 200-217 attempts to identify potential sources of uncertainty. However, validating the DGVM-based estimates with actual observed data could lend more credibility to this study.

Response: Concerning the comparison to Ref. 35 (Bultan et al., 2022) now in lines 295ff: There are substantial differences between our approach and the approach by Bultan et al. (2022). First, Bultan et al. (2022) assimilated observation-based carbon densities of woody vegetation (i.e., only forest and shrub PFTs) into the BLUE model, whereas we scale the BLUE vegetation and soil carbon densities for all PFTs based on carbon density ratios from DGVMs. As a consequence, our results cannot be compared directly. Second, Bultan et al. (2022) discuss that a large fraction of the 1.4 PgC yr⁻¹ increase can be attributed to high wood harvest and high deforestation rates in some regions (e.g., Southeast Asia, Equatorial Africa) imposed by the land-use forcing (LUH2) on the BLUE model, which does not match the biomass changes in the observational dataset in those regions. However, due to the direct assimilation of carbon densities in their approach, these mismatches led to a substantial increase in E_{LUC} . Bultan et al. (2022) discuss this in the context of the ‘assimilation bias’. Consequently, it is to be expected that the increase in E_{LUC} by including transient carbon densities would be lower (and consequently more in line with our results) if the LUH2 forcing data would be closer to the observed biomass changes. However, our results still agree with Bultan et al. (2022) in the sense that we find a non-negligible increase in E_{LUC} by accounting for transiently changing carbon densities, an important finding that is currently not considered in any carbon budgeting approach. We updated the text comparing our findings with Bultan et al. (2022) accordingly:

“Further, ref.³⁵ estimated an increase by 1.4 GtC yr⁻¹ in 2000–2019 by integrating an observation-based time series of woody vegetation carbon densities into BLUE compared to a simulation with static environmental conditions of 2000. This increase is substantially larger than our corresponding estimate of 0.2 (0.1, 0.3) GtC yr⁻¹ in 2000-2019. However, a direct comparison to our approach is difficult, since ref.³⁵ assimilates the absolute (observed) carbon densities in BLUE, whereas we use the trends from the DGVMs to scale the BLUE carbon densities. Consequently, discrepancies between the forcing dataset LUH2 and the observed carbon density dataset used in ref.³⁵ may have a larger impact on $E_{LUC,trans}$ compared to our approach.” (lines 300-307)

Concerning the comparison to observations: We have extended Table 2 (formerly Table 1) by observation-based estimates of the net land flux from O₂ measurements and from atmospheric inversions for the recent period and from atmospheric CO₂ and δ¹³C records for the cumulative net land flux. We are aware that also other estimates exist that aim at quantifying terrestrial carbon fluxes. However, comparison with these further estimates is not directly possible due to systematic and conceptual differences between our estimates and the observation-based estimates. First, as discussed in the manuscript, observations cannot separate S_{LAND} and E_{LUC} without further knowledge and/or assumptions on the underlying drivers (i.e., anthropogenic or natural), which makes modeling approaches, like bookkeeping models or DGVMs, essential to separate S_{LAND} and E_{LUC} . Currently, there is thus no observational dataset available that can directly provide separate estimates for S_{LAND} and E_{LUC} on a global scale. Second, Earth Observations can provide estimates of the net land flux (i.e. S_{LAND} plus E_{LUC}), but they do not include all compartments of the global carbon cycle. Harris et al. (2021, <https://doi.org/10.1038/s41558-020-00976-6>) estimated carbon fluxes in forests, thus missing

carbon fluxes from non-forested regions. Xu et al. (2021, <https://doi.org/10.1126/sciadv.abe9829>) estimated carbon fluxes from live biomass, thus missing carbon fluxes from soils. As our study includes carbon fluxes from forests and non-forested regions as well as from biomass and soils, we cannot directly compare our findings with the Earth Observation-based estimates from Harris et al. (2021) or Xu et al. (2021). Even if we did, we would not be able to attribute differences to potential shortcomings of our approach or to the systematic differences between the approaches. A third option for comparison would be upscaled data from FLUXNET towers (FLUXCOM initiative), as suggested by Reviewer 2. However, in this case a direct comparison is also not possible, as FLUXCOM estimates are based on upscaled fluxes of net ecosystem productivity (NEP), which do not necessarily include land use changes (e.g. Zscheischler et al., 2017; <https://doi.org/10.5194/bg-14-3685-2017>). In contrast, the DGVMs and bookkeeping models estimate net biome productivity (NBP), which includes land-use changes. Thus, mismatches in the FLUXCOM and BLUE estimates can again not directly be attributed to potential shortcomings of our approach or to systematic differences that arise from comparing NBP to NEP. Despite all these issues with comparisons to observation-based data, the available data for net land flux (based on O₂ measurements, atmospheric inversions, and atmospheric CO₂ and δ¹³C records) agree very well with our net land flux estimates (see Table 2) and are thus a valuable addition for our study.

The paper 'Modelled land use and land cover change emissions – a spatio-temporal comparison of different approaches' by the last author emphasizes an approach that provides a more robust accounting of fLULCC, for example, by estimating a mean DGVM ensemble fLULCC and LASC for a defined reference period and under homogeneous environmental changes (CO₂ only). It would be interesting to understand the differences between the approach used in these two studies. If they are not fundamentally different, it could question the novelty of this study."

Response: Obermeier et al. 2021 quantify E_{LUC} and the LASC using DGVM simulations from TRENDY. As we explain in our manuscript, it is not possible to split LASC into RSS and the environmental contribution to E_{LUC} with the current setup for DGVM simulations (see lines 62-66 and lines 232-238). Consequently, E_{LUC,trans} cannot be estimated by DGVMs. In contrast, we can estimate E_{LUC,trans} with our improvements to the bookkeeping model BLUE. To make this clearer, we added a section to the methods, which describes and explains the DGVM estimates of the terms in the terrestrial carbon budget (lines 616-629).

DGVMs are very important and informative for understanding the carbon cycles in global ecosystems; however, to accurately benchmark the global carbon budget, we should integrate more real observed data into these models.

Response: We fully agree with the reviewer that more model-data integration will be beneficial. The GCB and related projects work on the continuous improvement of their estimates. This includes that DGVMs are routinely evaluated using ILAMB in the GCB. There are also plans to integrate more observation-based information, e.g. on plant productivity or leaf area, into GCB models via data assimilation, or prescribe more information on specific processes, e.g. replace interactively simulated burned area by observational data. However, this is not yet integrated into the current operational, multi-model system of GCB, and is not free of errors itself (due to observational errors or again due to the fact that natural and anthropogenic effects cannot clearly be separated in observations). Our

approach will be able to pick up DGVM estimates that are better informed by observations as soon as they become available.

Author Response to Reviewer #2:

The inconsistent of CO₂ emissions from land-use change (ELUC) and of the terrestrial CO₂ sink significantly affects evaluation of carbon neutrality. Dorgeist et al. developed a new model named BLUE to precisely quantify the budgeting of terrestrial carbon fluxes. Generally, the topic is important and the scientific question is meaningful. The paper structure and the writing is good. However, based on my personal evaluation, I have several concerns for the BLUE model and the results.

Response: We thank the Reviewer for the constructive feedback and appreciate the valuable recommendations. We revised the manuscript accordingly as described in the following. We would like to emphasize that the BLUE model was developed and evaluated by Hansis et al. (2015, <https://doi.org/10.1002/2014GB004997>) and has since been used routinely in the GCB in its original set-up to deliver an estimate for E_{LUC} . It is also widely used in other studies focusing on global and regional land-use change fluxes (e.g. Schwingshackl et al., 2022, <https://doi.org/10.1016/j.oneear.2022.11.009>, Hong et al., 2022, <https://doi.org/10.1126/science.abj1572>). Our study improves and expands this existing model framework to not only deliver a more realistic E_{LUC} estimate, but also to estimate S_{LAND} , LASC and RSS. We made the routinely usage of BLUE more explicit in our manuscript:

“We derive transient carbon densities for vegetation and soil from DGVMs (responding to changes in environmental conditions) and implement them into the Bookkeeping of Land Use Emissions model⁷ (BLUE), one of three bookkeeping models routinely used in the GCB to deliver E_{LUC} estimates¹⁷ (lines 71-73)

Main comments

#1 Is the Imbalance of the global carbon budget in the BLUE affects the carbon evaluation results? To my personal understanding, Extended Data Fig. 1 showing that the imbalance of the global carbon budget in the BLUE is much higher than GCB and TRENDY. If the model is robust, the imbalance of BLUE should be around 0. However, the BLUE model seldom reach zero so an imbalance of BLUE may affect the terrestrial carbon fluxes evaluation. The carbon imbalance in BLUE may reach 1 Gt yr⁻¹ around 2005 to 2020. Compared with Fig.1 in the main text, the carbon flux from terrestrial ecosystems is only 0.3 Gt yr⁻¹, which is much less than the imbalance of BLUE (i.e. 1 Gt yr⁻¹). Therefore, whether the imbalance of BLUE affect the results, I have no idea about this.

Response: The budget imbalance (B_{IM}) is the residual of the two anthropogenic emissions terms (E_{LUC} and fossil emissions) and the natural sinks (atmosphere, ocean, S_{LAND}). It provides a measure of the discrepancies among the nearly independent estimates (Friedlingstein et al., 2023, <https://doi.org/10.5194/essd-15-5301-2023>). It gives a first-order plausibility check if the community understands the carbon cycle dynamics sufficiently well. However, it is known that a budget imbalance close to zero may be due to compensating errors (Friedlingstein et al., 2023, p. 5344). Various unresolved, known issues of the budget terms are discussed there, and unknown errors are expected. Suspected errors include, for example, that DGVMs do not capture the effects of diffuse light (O’Sullivan et al., 2021, <https://doi.org/10.1088/1748-9326/ac3b77>), or erroneous decadal variability due to DGVMs being oversensitive to wet decades, but also a range of issues around the

ocean sink, summarized in Friedlingstein et al. (2023). For the ocean sink, global (ocean) models and observational data products differ substantially; if the ocean sink estimate were based on the observational products instead of on models, the near-zero budget imbalance in the GCB would turn into a negative budget imbalance of more than -1 GtC yr^{-1} , indicating that a closure of the budget could only be achieved with anthropogenic emissions being significantly larger and/or the net land sink being substantially smaller than estimated in the GCB2023 - in line with what our study shows may indeed be the case! (However, we note again that many other compensating errors may be included in the B_{IM} .)

That the environmental contribution quantified in our study is on the order of 0.3 GtC yr^{-1} towards the end of our time series, as the reviewer notes, contributes to the BIM being more positive (which could be compensated if observational ocean products were used). That the BIM is as large as 1 GtC yr^{-1} in some years is attributable to additional effects contributing to interannual or decadal variability, like volcanic eruptions, wet/dry decades, or quasi-decadal ocean variability (Friedlingstein et al., 2023).

We have expanded the discussion on the B_{IM} in the manuscript to clarify the issues raised by the reviewer in lines 245-256 as follows:

“We note that our improvements to the terrestrial carbon budget do not bring the budget imbalance to zero - this cannot be expected due to many (potentially compensating) errors that accumulate in the imbalance term as a consequence of uncertainties in each of the five budget terms. Our results instead suggest a positive budget imbalance, which means that the estimated carbon sources are larger than the estimated carbon sinks. This imbalance could be explained by biases in other terms of the carbon budget. Indeed, a recent discussion suggests, among other potential causes for the imbalance, that estimates of the ocean sink in the GCB would be larger if they were based on the observation-based estimates of fugacity of CO_2 instead of being based on global ocean biogeochemistry models, as is the current GCB approach¹. A larger ocean sink is also supported by atmospheric inversion estimates¹ and would be consistent with the improved E_{LUC} and S_{LAND} estimates proposed by our study. We thus expect that improvements in other parts of the carbon budget would bring the budget imbalance closer to zero again.”

#2 Is there any ground surveying data or independent data source of carbon fluxes can prove that the BLUE model is robust?

This study only compared the data from simulation (i.e. GCB2022, TRENDY, BLUE, OSCAR). The critical question is all of these data are just from ‘model world’ but no evidence showing this model can reflect the ‘real world’. One possible solution is the authors may apply the ground surveying data such as carbon fluxes site data or the NEP data in FLUXCOM data to show whether the BLUE model can reflect the real world.

Response: As explained in response to reviewer 1, we have extended Table 2 (formerly Table 1) by observation-based estimates of the net land flux from O_2 measurements, from atmospheric inversions, and from atmospheric CO_2 and $\delta^{13}\text{C}$ records. We are aware that also other estimates exist that aim at quantifying terrestrial carbon fluxes. However, comparison with further estimates is not easily possible due to systematic and conceptual differences between our estimates and the observation-based estimates. First, as discussed in the manuscript, observations cannot directly separate S_{LAND} and E_{LUC} without further knowledge and/or assumptions on the underlying drivers (i.e.,

anthropogenic or natural), which makes modeling approaches, like bookkeeping models or DGVMs, essential to separate S_{LAND} and E_{LUC} . Currently, there is thus no observational dataset available that can provide separate estimates for S_{LAND} and E_{LUC} on a global scale. Second, Earth observations can provide estimates of the net land flux (i.e. S_{LAND} plus E_{LUC}), but they do not include all compartments of the global carbon cycle. Harris et al. (2021, <https://doi.org/10.1038/s41558-020-00976-6>) estimated carbon fluxes in forests, thus missing carbon fluxes from non-forested regions. Xu et al. (2021, <https://doi.org/10.1126/sciadv.abe9829>) estimated carbon fluxes from live biomass, thus missing carbon fluxes from soils. As our study includes carbon fluxes from forests and non-forested regions as well as from biomass and soils, we cannot directly compare our findings to the Earth Observation-based estimates from Harris et al. (2021) or Xu et al. (2021). Even if we did, we would not be able to attribute differences to potential shortcomings of our approach or to the systematic differences between the approaches. A third option for comparison would be upscaled data from FLUXNET towers (FLUXCOM initiative), as suggested by the Reviewer. However, in this case a direct comparison is also not possible, as FLUXCOM estimates are based on upscaled fluxes of local measurements of net ecosystem productivity (NEP), which do not necessarily include land use changes (e.g. Zscheischler et al., 2017; <https://doi.org/10.5194/bg-14-3685-2017>). In contrast, the DGVMs and bookkeeping models estimate net biome productivity (NBP), which includes land-use changes. Thus, mismatches in the FLUXCOM and BLUE estimates can again not directly be attributed to potential shortcomings of our approach or to systematic differences that arise from comparing NBP to NEP data. While all these issues with comparisons to observation-based data limit the number of observation-based studies that we could usefully include, the available data for the observation-based net land flux (based on O_2 measurements, atmospheric inversions, and atmospheric CO_2 and $\delta^{13}C$) agree very well with our net land flux estimates (see Table 2) and are thus a valuable addition for our study.

#3 The figures are nice but sometimes hard to read.

As this study is quantifying the carbon sink and source, I suggest all the figures can add a sign for whether the carbon fluxes represent for a carbon sink or source. For example, the Fig.1C, Fig.2, Fig.4 can add a sign for whether it is a carbon source or sink for positive and negative value of carbon fluxes.

Response: Thank you for pointing this out. We have added the suggested signs in Fig. 1a-c, Fig. 2, Fig. 4, and Supplementary Fig. 5.

Specific comments

L7 'We find that state-of-the-art process-based models overestimate S_{LAND} by 23% in 2012-2021 as they include hypothetical sinks that in reality are lost through historic ecosystem degradation.' So the overestimation of S_{LAND} will lead to what kind of results to the carbon neutrality? The authors may revise this sentence.

L9 'Additionally, E_{LUC} increases by 14% in 2012-2021 when considering environmental effects.' This also need to be revised. The increases of E_{LUC} will delay or accelerate the carbon neutrality?

Response: We included the following sentence in the abstract to indicate the impacts of the increased E_{LUC} and decreased S_{LAND} on carbon neutrality:

“Altogether, we find a weaker net land sink, which makes reaching carbon neutrality even more ambitious.” (lines 18-19)

Fig.1 One of the critical question is, why the author showing the BLUE output only from 2012 to 2021 in subplot b and c? Since the BLUE model can be extended from 1850 to present. Are these study years unique? Will it affect the carbon fluxes evaluation? For example, if the study years are from 1980 to 2021 or 2001 to 2021, will the CO₂ fluxes being quite different (Fig.1b)? For more results, the authors may consider adding a new figure or table in the SI.

Table1. This is another main concern for me. The BLUE model can express the E_{LUC} , SLAND, Net land flux from 1850 to 2021. But why the comparison of BLUE to the existed model or results are not consist? The BLUE output can also match the years (2009-2018) in Obermeier et al. with just adding a new line for their results. So one solution is the authors can focus on a certain time range (2012-2021) and highlight (one or two sentences at the introduction) whether they just compared the carbon fluxes at these years.

Response: Thank you for pointing this out. As suggested, we added estimates for further time periods in Supplementary Table 1 and show the split into components from Figure 1b for more time periods in Supplementary Fig. 5. We find that extending the time period back to 1992 (1992-2021) alters our flux estimates by not more than 0.2 GtC yr⁻¹. Notably, the BLUE estimates for 2009-2018, the period which is used by the estimates of Obermeier et al. (2021) and Gasser et al. (2020) in Table 2, only change very slightly compared to the BLUE estimates in 2012-2021. Thus, our choice of the study period 2012-2021 does not strongly affect our implications. Throughout the manuscript, we have included references to the estimates in Supplementary Table 1 when comparing to estimates from other studies that use a different period than 2012-2021.

Fig.4 When compared this results with Extended Fig.1, I am doubting about whether the imbalance will significantly affect the output of BLUE. If the imbalance or called as the uncertainties (Extended Fig.1) adding in Fig.4, what will the BLUE series results look like?

As I cannot evaluate whether the model is robust, I cannot give another more comments for the specific number or conclusions showing in the main text.

Response: We are not quite sure we understand the comment. We believe, however, that our above response on the budget imbalance and the additions to the manuscript clarify that the budget imbalance is determined by all terms in the global carbon budget and their errors. The shift of the budget imbalance using our improved BLUE model may in fact be consistent with errors suspected, e.g., on the ocean sink side (see comment above).

Here are also some suggestion for improving the methods.

L289 why choosing 8 models from TRENDY and called them as TRENDY output? The GCB2022 (TRENDY v11) also content these 8 models. The question is, why separated the TRENDY v11 into two results (GCB, TRENDY)?

Response: There are distinct differences between the fluxes that we label as “GCB2022” versus “TRENDY” in our study, which we clarify in the following, although there is an overlap, as the reviewer points out. Depending on the flux term, the GCB2022 uses either the average from three

bookkeeping models (E_{LUC}), the average from the TRENDY v11 ensemble (S_{LAND}) or a combination of both (Net land flux, see Table 2). This means that the GCB2022 is not restricted to the TRENDY models. In contrast to this, the fluxes that we label as 'TRENDY' always refer to the TRENDY DGVM ensemble, for E_{LUC} , S_{LAND} , as well as the net land flux. We have added a sentence to the Table 2 caption explaining that for S_{LAND} , TRENDY and GCB2022 estimates are identical.

L366 The improvement of RSS is important. Can the authors prove a full paragraph or a new figure on it? I also suggest the authors adding a concept figure for what is input and output of BLUE model and what is the improvement of BLUE to overcome the previous shortcomings. So a concept figure at the SI will be an alternative.

Response: Concerning the importance of RSS, we would like to point to the full paragraphs on RSS in our study, now in lines 166-175 and lines 192-203, and our introduction of RSS in lines 57-63 in the introduction. We tried to improve the clarity of both paragraphs to better highlight the importance of RSS. The RSS term is now additionally explained in (the new) Table 1.

We thank the reviewer for the suggestion to include a conceptual figure for the improvements of the BLUE model. We added such a figure in the supplementary (Supplementary Fig. 1) to visualize the steps of our analysis with BLUE and to emphasize the shortcomings of the current GCB approach (i.e., S_{LAND} being based on pre-industrial land cover and E_{LUC} excluding environmental effects).

Author Response to Reviewer #3:

Overview

The authors update a land use change bookkeeping model with transient environmental conditions from the TRENDY ensemble. In the Global Carbon Budget, land use change emissions and the natural carbon sink are estimated independently and therefore are incompatible. Through incorporating the effects of climate change on land use change and vice-versa, the authors claim that the terrestrial carbon sink is largely overestimated. They suggest that the GCB does not consider historic ecosystem degradation or the effect of environmental changes (like CO₂ fertilization) that increase the land sink and thereby increase CO₂ emissions under deforestation. This is a challenging effort, and I congratulate the authors on undertaking this work.

In general, this paper may be an interesting step forward in bridging separate modeling approaches of the terrestrial carbon sink and anthropogenic interventions. However, I have several major concerns that the authors must address before publication.

Response: We thank the Reviewer for the constructive feedback and appreciate the valuable recommendations, which we believe improve the quality of this manuscript. We revised the manuscript accordingly as described in the following.

Major comments

- I find the magnitude of the difference between TRENDY and the BLUE output to be quite astonishing (-9 GtC from GCB, vs. +53 GtC in current study) (line 167). These values do not align with estimates of the land carbon sink from atmospheric carbon isotopes, atmospheric inversions, or land sink estimates from biomass inventories (see Fig 1. Ruehr et al. 2023 Nature Reviews Earth & Environment). The authors do not thoroughly contextualize this result within the literature. Only two references are used in Table 1 to discuss the cumulative net land flux 1850-2018, which suggest widely different values of opposite sign than the authors report (although with large certainty ranges), and little explanation is given for large this divergence. Additional discussion and observational evidence in support of their claim is necessary.

Response: Thank you for pointing this out. We have extended Table 2 (formerly Table 1) by observation-based estimates of the net land flux from O₂ measurements and from atmospheric inversions for the recent period and from atmospheric CO₂ and δ¹³C records for the cumulative net land flux. We have contacted the authors of the Ruehr et al. (2023) study to better understand the data that they used. Their Figure 1 shows estimates for the natural land sink (S_{LAND}). The inversion-based estimates that they use are typically used to quantify the net land flux. Some inversion-based systems output an estimate for S_{LAND} by accounting for biomass burning as a proxy for land-use emissions (personal communication, Ingrid Lujikx). However, as biomass burning is only part of the land-use emissions and, e.g., neglects legacy emissions from the decay of biomass, we decided to only include the net land flux from atmospheric inversions, as that estimate does not require additional assumptions. The DGVM estimates included by Ruehr et al. (2023) are already considered in our study. The atmospheric CO₂ and δ¹³C records from Joos et al. (1999) are an estimate for the net land flux (like the inversion-based estimates) and we thus use them for the net land flux estimates rather than for S_{LAND} . These records only cover the time until 1995 and can thus not be directly compared to our cumulative net land flux from 1850-2021. We thus added a BLUE estimate

of the cumulative net land flux for 1850-1995 in Supplementary Table 1. The BLUE estimate for this period yields larger net emissions than the Joos estimate, which indicates that the cumulative net land flux might indeed be close to zero in 1850-2021. However, given the large uncertainties of all these estimates, we refrain from giving a final answer to whether the cumulative net land flux has been a source or sink in 1850-2021:

“The cumulative net land flux over 1850-2021 from BLUE indicates that land has been a net source of CO₂ of 53 (-21, 117) GtC, while TRENDY estimates a small net CO₂ sink of -2 ± 112 GtC as does the GCB with -9 ± 137 GtC (Fig. 4, Table 2). The good agreement of the cumulative GCB estimate with TRENDY is likely due to compensating biases with GCB having larger net emissions than TRENDY before 1970 and larger net sinks afterwards (Fig. 4). Another estimate of the net land flux based on atmospheric CO₂ and $\delta^{13}\text{C}$ records yields 31 (-26, 88) GtC cumulatively in 1850-1995²³. For the same period, BLUE yields a cumulative net land flux of 78 (24, 123) GtC (Supplementary Table 1). As all these estimates bear large uncertainties, it remains inconclusive whether land has been a cumulative sink or source of CO₂ since 1850.” (lines 223-231)

We are aware that also other estimates exist that aim at quantifying terrestrial carbon fluxes. However, comparison with these further estimates is not directly possible due to systematic and conceptual differences between our estimates and the observation-based estimates. First, as discussed in the manuscript, observations cannot separate S_{LAND} and E_{LUC} without further knowledge and/or assumptions on the underlying drivers (i.e., anthropogenic or natural), which makes modeling approaches, like bookkeeping models or DGVMs, essential to separate S_{LAND} and E_{LUC} . Currently, there is thus no observational dataset available that can directly provide separate estimates for S_{LAND} and E_{LUC} on a global scale. Second, Earth Observations can provide estimates of the net land flux (i.e. S_{LAND} plus E_{LUC}), but they do not include all compartments of the global carbon cycle. Harris et al. (2021, <https://doi.org/10.1038/s41558-020-00976-6>) estimated carbon fluxes in forests, thus missing carbon fluxes from non-forested regions. Xu et al. (2021, <https://doi.org/10.1126/sciadv.abe9829>) estimated carbon fluxes from live biomass, thus missing carbon fluxes from soils. As our study includes carbon fluxes from forests and non-forested regions as well as from biomass and soils, we cannot directly compare our findings with the Earth Observation-based estimates from Harris et al. (2021) or Xu et al. (2021). Even if we did, we would not be able to attribute differences to potential shortcomings of our approach or to the systematic differences between the approaches. A third option for comparison would be upscaled data from FLUXNET towers (FLUXCOM initiative), as suggested by Reviewer 2. However, in this case a direct comparison is also not possible, as FLUXCOM estimates are based on upscaled fluxes of net ecosystem productivity (NEP), which do not necessarily include land use changes (e.g. Zscheischler et al., 2017; <https://doi.org/10.5194/bg-14-3685-2017>). In contrast, the DGVMs and bookkeeping models estimate net biome productivity (NBP), which includes land-use changes. Thus, mismatches in the FLUXCOM and BLUE estimates can again not directly be attributed to potential shortcomings of our approach or to systematic differences that arise from comparing NBP to NEP. Despite all these issues with comparisons to observation-based data, the available data for net land flux (based on O₂ measurements and atmospheric inversions) for the recent period agree very well with our net land flux estimates (see Table 2) and are thus a valuable addition for our study.

- In several places in the text, I found the writing unnecessarily confusing, with long sentences and lots of parenthetical comments. See line comments below for details. The introduction, results and

discussion should be edited to prioritize clarity. I recommend breaking apart long sentences into multiple sentences and avoid unnecessary complexity.

Response: Thank you for making us aware of this. Certainly, we agree that unnecessarily confusing should be avoided. We have tried to improve the clarity throughout the manuscript during our revisions. We revised the mentioned parts accordingly (see reviewer comments and our responses below).

- Consider changing the title. The current title is not related to your claim, which is that the terrestrial biosphere has been a net source of carbon since 1850 after accounting for dynamic environmental effects on land use change and vice-versa. The current title does not tell us much about your conclusions or why this paper is interesting.

Response: We prefer to stay with the original direction of title since the consistent approach to terrestrial carbon budgeting is overarching over changes in the estimates of individual terms, of which we discuss several. Additionally, we do not want to stress too much the fact that the terrestrial biosphere could have been a net source of carbon since 1850, as the uncertainties of these estimates are substantial (see manuscript and our answer to the inclusion of observation-based datasets above).

- Consistent labeling and providing definitions would benefit clarity. A table of definitions is needed -- LULUCF, LASC, RSS, Sland, Eluc, natural vs. net land sink, etc. Is Fig 4 Y axis ('net land flux) the same as Fig 3 shading of Fig 2 Y axis ('CO2 flux')? Why different units for panel C of Fig 1, if this too could be CO2 flux?

Response: We thank the reviewer for this suggestion. We provided a table of definitions for the most important terms within the terrestrial carbon budget in Table 1. To address the reviewer's concerns about the differences in labeling of the y-axis in Fig.1 - 4 and the different units in Fig. 1c, we unified all y-axis labels to "CO2 flux". However, for panel Fig. 1c, we decided to maintain "env. contrib." to highlight that the map shows the environmental contribution.

- I suggest additional focus on $E_{LUC,trans}$ in helping to clarify the text. Having pre-industrial, present-day, and transient conditions is interesting, but it is also more confusing. Consider instead focus on the difference between $E_{LUC,pi}$ or $E_{LUC,pd}$ and $E_{LUC,trans}$. You might also consider taking differences between $E_{LUC,pi}$ or $E_{LUC,pd}$ and $E_{LUC,trans}$ in Figs S4 and S5 to better visualize the difference over time, since it is so close in magnitude and hard to see.

Response: We understand that the comparison of $E_{LUC,pi}$ or $E_{LUC,pd}$ and $E_{LUC,trans}$ makes the content sometimes harder to follow, but we think that a comparison of $E_{LUC,trans}$ with both $E_{LUC,pi}$ and $E_{LUC,pd}$ is of importance. The comparison to both $E_{LUC,pi}$ and $E_{LUC,pd}$ makes it possible to show the impact of transient environmental conditions on the one hand (difference between $E_{LUC,trans}$ and $E_{LUC,pi}$), and the shortcomings of the current GCB estimate on the other (difference between $E_{LUC,trans}$ and $E_{LUC,pd}$). We added the difference of $E_{LUC,trans}$ and $E_{LUC,pi}$ (i.e., the environmental contribution to E_{LUC}) in Supplementary Fig. 7 to emphasize the difference between E_{LUC} estimates. Additionally, we increased the size of the figure to improve visibility of the differences over time.

Line comments

- Abstract: in your implications, add that the carbon sink is weaker (and perhaps a source since

1850) than previously thought. This overestimate of the land carbon sink may undermine efforts to limit CO₂ pollution.

Response: We included the following sentence in the abstract:

“Altogether, we find a weaker net land sink, which makes reaching carbon neutrality even more ambitious.” (lines 18-19)

- Lines 19-21: unclear wording. Do “changes” refer to trends in LULUCF and environment, or are these just average fluxes over the period listed?

Response: We have adjusted the sentence (now in lines 27-31) to clarify that changes refer to (mostly long-term) changes over history.

- Line 24: reword “separately indicate” to “uses separate methods to estimate”

Response: We revised this part of the introduction such that this sentence does not exist anymore in the manuscript.

- Line 40-44: split into multiple sentences, this is too long and confusing

Response: As suggested, we split the paragraph into multiple sentences (see lines 55-57).

- Line 50: Distinction between RSS and Eluc is confusing. Please provide further clarity or table of definitions.

Response: As suggested, we included the definitions of the terms of the terrestrial carbon budget in Table 1 which clarifies the distinction between E_{LUC} , RSS and LASC. Further, we extended the explanation of RSS and LASC in the introduction in lines 54-69 which reads as follows:

“[In contrast,] DGVMs consider transient environmental effects (effects changing over time) for estimating S_{LAND} . Historically, environmental effects, such as rising CO₂ levels, have been mainly beneficial for plant growth, in particular for forests with their long-lived woody biomass). As a consequence, S_{LAND} has been a carbon sink globally¹. However, by design of the simulation setup, DGVMs estimate S_{LAND} under pre-industrial land cover, thus including effects of environmental changes in forest areas that, in reality, have since been lost due to LULUCF. The hypothetical carbon sinks in these lost ecosystems are also known as replaced sinks and sources¹⁵ (RSS). The RSS term is of substantial size with 31 GtC of hypothetical sinks cumulatively from 1850-2018 (as estimated by ref.⁹ with a bookkeeping method). However, due to limitations in the simulation setup, RSS cannot be isolated directly with DGVMs (see Methods). In analyses based on DGVMs, RSS are always lumped together with the effect of environmental changes on E_{LUC} in a term summarized as loss of additional sink capacity¹⁶ (LASC). The LASC term combines carbon fluxes from environmental changes on land that have been altered due to LULUCF and from changes in E_{LUC} due to environmental effects (see Table 1). The missing environmental effects in the bookkeeping estimates of E_{LUC} and the assumption of a constant, pre-industrial land cover in the DGVM estimates of S_{LAND} currently prohibits closing the terrestrial carbon budget.”

- Line 79 onward: Provide information on what these changes are relative to

Response: We revised the paragraph such that it becomes clear that the observed changes in $E_{LUCtrans}$

(the simulation including environmental changes) are relative to $E_{LUC,pi}$ (the simulation excluding environmental changes). The paragraph (lines 107-111) now reads as:

“Upon deforestation and wood harvest, the higher carbon stocks of vegetation and soil increase CO₂ emissions in $E_{LUC,trans}$ compared to $E_{LUC,pi}$ by 24% (0.4 GtC yr⁻¹) and 22% (0.3 GtC yr⁻¹, Fig. 1b, Supplementary Figs. 5 and 6). These larger emissions are only partly compensated by increased sinks through re/afforestation (increase by 18%, 0.2 GtC yr⁻¹) and regrowth after wood harvest (increase by 17%, 0.1 GtC yr⁻¹).”

- Line 85: “This may...” Provide more implications. Over long time periods (and define scale of these), what might we expect the effect of environmental changes on Eluc to be?

Response: Realizing that this sentence was misleading and since it contained marginal information only, we have removed it. We discuss in lines 50-52 that environmental changes express themselves through, e.g., denser growing forests in response to a rising atmospheric CO₂ concentration and thus emit more when cleared for agricultural land. In the Discussion section we discuss impacts of future environmental changes, including weather extremes.

The effects on E_{LUC} are discussed in lines 333-326:

“As future environmental conditions are expected to diverge further from the present-day state, E_{LUC} estimates are projected to grow further apart in future decades⁹, yet decisively depending on the future evolution of CO₂ concentrations and climate change³⁷.”

The long term effects on S_{LAND} and RSS are described in lines 342-348:

“Losses in S_{LAND} due to LULUCF are expected to increase even further because halting deforestation and forest degradation is targeted to be achieved only by 2030⁴¹, thus further augmenting the lost sinks (i.e., RSS), which accumulate over time¹⁶ (Table 2). Additionally, damages from climate change, such as land drying, droughts, and wildfires, may increasingly counteract the beneficial effects of increased atmospheric CO₂ levels on plant growth^{30,42,43}. The long-term evolution of S_{LAND} and RSS thus crucially depends on our climate mitigation efforts and on which of the vastly different potential future land-use paths we follow⁴⁴.”

- Lines 79-92: This paragraph is very clear, primarily because it provides concrete examples. I would suggest moving this paragraph up to beginning of results, or providing some of these examples within your introduction to better set the stage.

Response: Thank you for your positive feedback about this paragraph. We prefer maintaining the structure in the results section as is. The mentioned paragraph elaborates on the component fluxes which we would like to include after showing our main results on E_{LUC} based on different environmental conditions. However, we want to stress that we put substantial effort into making the introduction clearer, also responding to your other comments.

- Line 88: replace “met” with “occurred within”

Response: The revised text reads as follows in lines 111-113: “Environmental impacts on E_{LUC} are largest in tropical regions (Fig 1c, Supplementary Fig. 7), where relatively recent and substantial forest clearings occurred under strongly increased carbon stocks.”

- Line 131: Break into two sentences. Put parentheses into a separate sentence.

Response: We revised the text as suggested in lines 170-172.

- Line 175: I am confused about the RSS and Eluc not being able to be separated. More information here would be useful

Response: DGVMs are able to quantify the LASC, but they are not able to separate the LASC into RSS and the environmental contribution to E_{LUC} . Such a separation is necessary to calculate $S_{LAND,trans}$ and $E_{LUC,trans}$. To make this clearer, we extended the sentence in lines 234-238 in the main text:

“Moreover, the split of the net land flux from TRENDY into E_{LUC} ($E_{LUC,TRENDY}$) and S_{LAND} ($S_{LAND,TRENDY}$) is confounded by LASC. With the current setup for DGVM simulations it is not possible to split LASC into RSS and the environmental contribution to E_{LUC} (see Methods), and DGVMs can thus not deliver all terms necessary for a holistic and consistent terrestrial carbon budget.”

Additionally, we included a new section in the Method called “DGVM estimates of the terms in the terrestrial carbon budget” explaining the shortcomings of the DGVM estimates (lines 616-629).

- Lines 188-195: reported changes are relative to the ELUC without climate effects, correct? Spell that out more clearly. Consider creating a figure focusing on this difference (ELCUC_pd vs. ELUC_trans) as this seems to be closer to your main conclusion.

Response: These sentences discuss the comparison of E_{LUC} and S_{LAND} estimates from BLUE that are conceptually similar to the E_{LUC} and S_{LAND} estimates from TRENDY. The idea is as follows: With BLUE, we can calculate an S_{LAND} estimate that is conceptually similar to the TRENDY estimate of S_{LAND} , i.e., assuming constant pre-industrial land cover ($S_{LAND,pi}$). Likewise, we can produce an ELUC estimate with BLUE that includes RSS, as the TRENDY estimate of ELUC does (see Eq. (10) in the Methods). To make this clearer, we revised the manuscript in lines 257-263 as follows:

“With our approach we are able to quantify all major components of the terrestrial carbon budget with BLUE: E_{LUC} , S_{LAND} , the environmental contribution to E_{LUC} , RSS, and LASC (as the sum of the latter two). This makes it possible to mimic other estimates of terrestrial carbon budget terms. For instance, the sum of $E_{LUC,trans}$ and RSS conceptually equals $E_{LUC,TRENDY}$ (see Methods), both showing an upward trend from the 1960s onwards due to the accumulating nature of LASC (Extended Data Fig. 2a). Similarly, we can deduce a BLUE estimate for S_{LAND} under pre-industrial land cover ($S_{LAND,pi}$) that conceptually agrees with $S_{LAND,TRENDY}$ (Extended Data Fig. 2b).”

- Move lines 230-233 to introduction. This is a good explanation in plain English that would benefit lay readers earlier in the text.

Response: Thank you for your positive feedback on these lines. We put substantial effort into making the introduction better understandable. In particular, we tried to make the motivation and structure of our study more apparent in the introduction. Nevertheless, we would like to keep this sentence here, as we think that it provides a good start into the discussion, which we also deem important.

- Similarly, lines 451-453 are a helpful definition that could be moved up to the intro or results sections.

Response: We included the respective definitions in Table 1.

Figure comments

- Fig 1: interesting that $E_{LUC,pd}$ and $E_{LUC,trans}$ are so similar until ~2000. Why?

Response: Thank you for pointing this out. We would rather say that $E_{LUC,pd}$ and $E_{LUC,trans}$ are similar in the decades around 1980. This can be explained as follows: First, $E_{LUC,trans}$ and $E_{LUC,pi}$ are similar until the end of the 19th century due to similar carbon densities, as changes in CO_2 and climate were still small back then. Afterwards, $E_{LUC,trans}$ approximates $E_{LUC,pd}$ as carbon densities become more similar. $E_{LUC,trans}$ eventually overtakes $E_{LUC,pd}$ in the 1980s as carbon densities become generally larger for all PFTs (see Supplementary Fig. 3 and 4). This is because the literature-based values of BLUE and previous bookkeeping models are representative for approximately the 1980s, which is why changes in environmental conditions were accounted for relative to 1980 (see Methods, section “BLUE simulations”). Additionally, differences between the three runs are not only dependent on the different carbon densities but also on land use changes happening (i.e., more land use changes after the 1950s) leading to larger emissions when larger carbon densities are considered.

- Fig 2: Unclear what black arrow refers to. Difference between BLUE and TRENDY? Clarify in caption. Also clarify blue arrow.

Response: $S_{LAND,trans}$ and $S_{LAND,pi}$ are derived from BLUE and differ in their respective considered land cover, i.e., the lost sinks due to historic ecosystem degradation. The black arrow (which we changed to red to improve clarity) shows the difference between $S_{LAND,pi}$ (from BLUE) and $S_{LAND,TRENDY}$ (from DGVMs). Here, the difference is not the considered land cover (they are both based on pre-industrial land cover), but the model-intrinsic differences between BLUE and DGVMs, e.g., distributions of natural vegetation types, and assumed carbon densities. We added this explanation in the caption of Figure 2 (“ $S_{LAND,pi}$ and $S_{LAND,TRENDY}$ are both based on pre-industrial land cover and thus, their difference is due to model-intrinsic differences between BLUE and the DGVMs from the TRENDY project.”).

- Fig S1: Mostly for my own curiosity – I see lines on country borders (between USA and Canada or PNG/Indonesia) where emissions from land-use change flip from negative to positive. Curious how country boundaries affect the model output. Perhaps include some information here. Also, why such strong but opposite trends in northern India / Bangladesh area?

Response: The land use change data (LUH2) that is used to force the TRENDY and BLUE models is based on country-level agricultural land use data from the FAO. The sharp changes at country borders in the mentioned regions are therefore imposed on the models via the forcing data. As discussed in Bultan et al. (2022), the strong trends in Northern India/Bangladesh are related to very high wood harvest rates from the early 2000s onwards. Unfortunately, there is no comparable observational dataset that could be used to validate the wood harvest dynamics. However, Bultan et al. (2022) find a strong mismatch between observed biomass changes and the simulated biomass changes in BLUE (driven by the high wood harvest rates from LUH2) in this region and therefore conclude that the high wood harvest rates might be unrealistic.

- Fig S3: hard to see differences between models as they are grouped. Consider adjusting y-axes to smaller range; the data are much smaller in range than the axes.

Response: As suggested by the Reviewer we adjusted the y-axis in Supplementary Fig. 3 and 4 to a smaller range to improve visibility of the differences between models.

- Fig S6: Scales on Y-axes change for each regional estimate. Consider adding an additional figure (box plot?) to more easily compare regions. You might show Sland,act cumulative for each region on one figure with same Y scale for relative differences.

Response: We considered unifying the y-axis of the single subpanels to make the regions better comparable, but in that case the temporal evolution would become almost invisible in regions with very small carbon fluxes. We thus decided to keep the figure as it is, and we additionally highlight in the caption that the scale of the y-axis differs across regions. Given that we already have many figures in the supplementary information, we decided against adding another figure.

REVIEWERS' COMMENTS

Reviewer #2 (Remarks to the Author):

Thank you for the authors' effort.

The current manuscript is almost reach the quality for publication.

I just have one more suggestion. As reviewer 1 and me have raised the concern about validating the DGVM-based estimates with actual observed data. The explanation from authors are acceptable and I recommend to shorten this paragraph into the discussion section, which can be part of the uncertainties explanation of the current study. Sometimes point out the uncertainty is kind of accuracy to the science community.

Reviewer #3 (Remarks to the Author):

The authors have done a thorough job responding to my comments and those of the other reviewers. I congratulate them especially on their efforts to improve the manuscript's clarity in the introduction, which reads well, and to better contextualize their results within the literature.